# Sample Efficient Reinforcement Learning
# via Low-Rank Matrix Estimation

**Devavrat Shah**
EECS, MIT
devavrat@mit.edu

**Dogyoon Song**
EECS, MIT
dgsong@mit.edu

**Zhi Xu**
EECS, MIT
zhixu@mit.edu

**Yuzhe Yang**[*]
EECS, MIT
yuzhe@mit.edu

## Abstract

We consider the question of learning $Q$-function in a sample efficient manner for reinforcement learning with continuous state and action spaces under a generative model. If $Q$-function is Lipschitz continuous, then the minimal sample complexity for estimating $\epsilon$-optimal $Q$-function is known to scale as $\Omega(\frac{1}{\epsilon^{d_1+d_2+2}})$ per classical non-parametric learning theory, where $d_1$ and $d_2$ denote the dimensions of the state and action spaces respectively. The $Q$-function, when viewed as a kernel, induces a Hilbert-Schmidt operator and hence possesses square-summable spectrum. This motivates us to consider a parametric class of $Q$-functions parameterized by its "rank" $r$, which contains all Lipschitz $Q$-functions as $r \to \infty$. As our key contribution, we develop a simple, iterative learning algorithm that finds $\epsilon$-optimal $Q$-function with sample complexity of $\widetilde{O}(\frac{1}{\epsilon^{\max(d_1,d_2)+2}})$ when the optimal $Q$-function has low rank $r$ and the discounting factor $\gamma$ is below a certain threshold. Thus, this provides an exponential improvement in sample complexity. To enable our result, we develop a novel Matrix Estimation algorithm that faithfully estimates an unknown low-rank matrix in the $\ell_\infty$ sense even in the presence of arbitrary bounded noise, which might be of interest in its own right. Empirical results on several stochastic control tasks confirm the efficacy of our "low-rank" algorithms.

## 1 Introduction

Reinforcement Learning (RL) has emerged as a promising technique for a variety of decision-making tasks, highlighted by impressive successes in solving Atari games [29, 30] and Go [39, 40]. However, generic RL methods suffer from "curse-of-dimensionality." Specifically, the classical minimax theory [41, 45] suggests that for $\epsilon > 0$, we need $\Omega(\frac{1}{\epsilon^{d_1+d_2+2}})$ samples to learn an $\epsilon$-optimal state-action value, i.e., $Q$-function, when the (continuous) state and action spaces have dimensions $d_1$ and $d_2$ respectively and the $Q$-function is Lipschitz continuous. On the other hand, as exemplified by empirical successes, practical RL tasks seem to possess low-dimensional latent structures. Indeed, feature-based methods aim to explain such a phenomenon by positing that either the transition kernel [48, 49] or the value function [44, 28, 31, 26, 52] is linear in low-dimensional features associated with states and actions. That is, not only the states and actions have low-dimensional representations, the value function is linear. While these may be true, the algorithm may not have the *knowledge* of such feature maps beforehand; and relying on the hope of a neural network to find them might be too much to ask.

With these motivations, the primary goal of this work is to learn the optimal $Q$-function in a data-efficient manner when it has a lower-dimensional structure, *without* the need for any additional information such as knowledge of features. Thus, we ask the following key question in this paper:

> "*Is there a universal representation of Q-function that allows for designing a data-efficient learning algorithm if the Q-function has a low-dimensional structure?*"

---

[*]The author ordering is alphabetical.

Table 1: Informal summary of sample complexity results for three different configurations of state/action spaces, including our results, a few selected from the literature, and the lower bounds. See Them. 2 & Appx. E for ours.

| Setting | Our Results | Selected from the Literature | | Lower Bound |
|---------|-------------|---------------|---------------|-------------|
| Cont. $\mathcal{S}$ & Cont. $\mathcal{A}$ | $\tilde{O}\big(\frac{1}{\epsilon^{\max\{d_1,d_2\}+2}}\big)$ | N/A | | $\Omega\big(\frac{1}{\epsilon^{d_1+d_2+2}}\big)$ [45] |
| Cont. $\mathcal{S}$ & Finite $\mathcal{A}$ | $\tilde{O}\big(\frac{1}{\epsilon^{d_1+2}}\big)$ | $\tilde{O}\big(\frac{1}{\epsilon^{d_1+3}}\big)$ [35] | $\tilde{O}\big(\frac{1}{\epsilon^{d_1+2}}\big)$ [51] | $\tilde{\Omega}\big(\frac{1}{\epsilon^{d_1+2}}\big)$ [35] |
| Finite $\mathcal{S}$ & Finite $\mathcal{A}$ | $\tilde{O}\big(\frac{\max(|\mathcal{S}|,|\mathcal{A}|)}{\epsilon^2}\big)$ | $\tilde{O}\big(\frac{|\mathcal{S}||\mathcal{A}|}{(1-\gamma)^3\epsilon^2}\big)$ [37] | $\tilde{O}\big(\frac{|\mathcal{S}||\mathcal{A}|}{(1-\gamma)^4\epsilon^2}\big)$ [38] | $\tilde{\Omega}\big(\frac{|\mathcal{S}||\mathcal{A}|}{(1-\gamma)^3\epsilon^2}\big)$ [3] |

**Contributions.** As the main contribution of this work, we answer this question in the affirmative by developing a novel spectral representation of the $Q$-function for a generic RL task, and provide a data-efficient method to learn a near-optimal $Q$-function when it is lower-dimensional.

*Representation.* Given state space $\mathcal{S} = [-1,1]^{d_1}$ and action space $\mathcal{A} = [-1,1]^{d_2}$, let $Q^* : \mathcal{S} \times \mathcal{A} \to \mathbb{R}$ be the optimal $Q$-function for the RL task of interest. We consider the integral operator $K = K_{Q^*}$ induced by $Q^*$ that maps an integrable function $h : \mathcal{S} \to \mathbb{R}$ to $Kh : \mathcal{A} \to \mathbb{R}$ such that $Kh(a) = \int_{s\in\mathcal{S}} Q^*(s,a)h(s)ds$, $\forall a \in \mathcal{A}$. For Lipschitz $Q^*$, we show that $K$ is a Hilbert-Schmidt operator admitting generalized singular value decomposition. This leads to the spectral representation of $Q^*$:

$$Q^*(s,a) = \sum_{i=1}^{\infty} \sigma_i f_i(s)g_i(a), \quad \forall s \in \mathcal{S}, a \in \mathcal{A}, \tag{1}$$

with $\sum_{i=1}^{\infty} \sigma_i^2 < \infty$, and "singular vectors" $\{f_i : i \in \mathbb{N}\}$ and $\{g_i : i \in \mathbb{N}\}$ being orthonormal sets of functions. That is, for any $\delta > 0$, there exists $r(\delta)$ such that the $r(\delta)$ components in (1) provide $\delta$-approximation of $Q^*$. This inspires a parametric family of $Q^*$ parameterized by $r \geq 1$, i.e., $Q^*(s,a) = \sum_{i=1}^{r} \sigma_i f_i(s)g_i(a)$, with all Lipschitz $Q^*$ captured as $r \to \infty$. When $r$ is small, it suggests a form of lower-dimensional structure within $Q^*$: we call such a $Q^*$ to have *rank $r$*.

*Sample-Efficient RL.* Given the above universal representation with the notion of dimensionality for $Q^*$ through its rank, we develop a data-efficient RL method. Specifically, for any $\epsilon > 0$, our method finds $\hat{Q}$ such that $\|\hat{Q} - Q^*\|_\infty < \epsilon$ using $\tilde{O}\big(\epsilon^{-(\max\{d_1,d_2\}+2)}\big)$ samples, with the hidden constant in $\tilde{O}(\cdot)$ dependent on $r, \max\{d_1,d_2\}$ (cf. Theorem 2). In contrast, the minimax lower bound for learning a generic Lipschitz $Q^*$ in the $L^\infty$ sense (also in the $L^2$-sense) is of $\Omega\big(\epsilon^{-(d_1+d_2+2)}\big)$ [45]. That is, our method removes the dependence on the smaller of the two dimensions by exploiting the low-rank structure in $Q^*$. Note that this provides an exponential improvement in sample complexity, e.g., with $d_1 = d_2 = d$, our method requires the number of samples scaling as $\epsilon^{-d-2}$ in contrast to $\epsilon^{-2d-2}$ required for generic Lipschitz $Q^*$. For a quick comparison, see Table 1 and Related Work.

*Matrix Estimation (ME), A Novel Method.* Our data-efficient RL method relies on a novel low-rank Matrix Estimation method we introduce. Notice that for any set of $m$ states $\{s_k\}_{k=1}^m$ and $n$ actions $\{a_\ell\}_{\ell=1}^n$, the induced matrix $[Q^*(s_k,a_\ell) : k \in [m], \ell \in [n]]$ has rank (at most) $r$. At a high level, to obtain the improved sample complexity as claimed, we wish to faithfully recover the $m \times n$ rank-$r$ matrix in the $\ell_\infty$ sense, by observing only $\tilde{O}\big(\max(m,n)r\big)$ entries with each entry having bounded, but arbitrary noise $\delta$. In the literature [8, 9, 11, 15], such a harsh setting has not been considered. In this work, we introduce a ME method that manages to recover the entire matrix with entry-wise error within $O(\delta)$ (cf. Proposition 5). This advance in ME should be of independent interest (see Table 2 for comparison). With this novel method, we improve our estimates of $Q^*$ iteratively by interleaving one-step lookahead and ME. This, ultimately leads to an $\epsilon$-optimal $Q^*$ with desired sample size.

*Empirical Success.* While low-rank representation of $Q^*$ enables theoretical guarantees, the proof is in the puddling: we find that for well-known control tasks, the underlying $Q^*$ has a low-rank structure. In particular, using our method that exploits the low-rank structure leads to a significant improvement in sample complexity over the method that does not. Our novel ME, with provable guarantees, turns out to be computationally most efficient, while offering superior performance of sample complexity.

*Summary.* To the best of our knowledge, this is the first work to show a provable, quantitative utility of exploiting the low-rank structure to reduce sample complexity in $Q$-learning. We believe that "factorization" of $Q^*$ can be beneficial more broadly in improving the efficiency of RL, e.g., it could be embedded as an architectural constraint in neural network representation of the $Q^*$. Moreover, the main insight we develop in this paper remains valid and applicable to various problems in machine learning beyond RL, which involve a bivariate function possessing a low-rank structure.

Table 2: Comparison of different ME methods with different guarantees. Ours is the only method that provides entry-wise guarantee while allowing for arbitrary, bounded error in each entry.

| Method | Noise Model | Error Guarantees | Sampling Model | # of Samples |
|---|---|---|---|---|
| Our Method | bounded arbitrary | entrywise | adaptive | $O(n)$ |
| Convex Relaxation | noiseless | exact | independent w.p. $p$ | $O(n \log^2 n)$ |
| [9, 7, 23] | bounded arbitrary | Frobenius | independent w.p. $p$ | $O(n \log^2 n)$ |
| Spectral Thresholding [10] | zero-mean | Frobenius | independent w.p. $p$ | $O(n^{1+c})$ |
| Factorization (noncvx) [13] | zero-mean | entrywise | independent w.p. $p$ | $O(n \log^3 n)$ |

**Related Work.** A brief discussion of related work on RL and Matrix Estimation is provided.

RL problems with both continuous state and action spaces received significantly less attention in the literature. While there are practical RL algorithms to deal with continuous domains [46, 24, 20, 25], theoretical understanding on this class of problems, especially on sample complexity, is very limited [1]. Since we interpolate our estimates to the entire space via non-parametric regression without making any additional model assumptions, a comparison with the non-parametric minimax rate $\Omega(\frac{1}{\epsilon^{d_1+d_2+2}})$ for learning Lipschitz function [41, 45] is meaningful.

Our algorithm and proofs are general, which can be reduced to low-rank settings with a finite (discrete) space in a similar manner (Appendix E.3). The lower bound scales as $\tilde{\Omega}(\frac{1}{\epsilon^{d+2}})$ for problems with continuous state space and finite action space [35] and $\tilde{O}(\frac{|\mathcal{S}||\mathcal{A}|}{(1-\gamma)^3\epsilon^2})$ for problems with both state and action spaces being finite [3]. When reduced to those domains, our method scales as $\tilde{O}(\frac{1}{\epsilon^{d+2}})$ for the former and $\tilde{O}(\frac{\max(|\mathcal{S}|,|\mathcal{A}|)}{\epsilon^2})$ for the latter, respectively. That is, the smaller of the two dimensions is "removed" from sample complexity by exploiting the low-rank structure in the same way as in the continuous problems. Results in finite domains are abundant in the literature and it is impossible to cover them all. We provide a high-level summary in Table 1 to communicate how our algorithm fares with a few selected work. Note that the detailed setting often varies in the literature and we refer readers to Appendix G for further discussions. Finally, we remark that our analysis requires the discounting factor $\gamma$ to be small, and leave it as an important future direction to extend to all $\gamma$.

We mention the recent empirical work [50] that investigates low-rank $Q^*$ with matrix estimation for finite state and action spaces. The results in [50] are solely empirical and it uses off-the-shelf ME methods. In that sense, we provide a formal framework to understand why [50] works so well, resolving the theoretical open problem raised in their work, and we provide natural generalization for continuous state and action spaces that was missing, along with a novel ME method.

As discussed, matrix estimation concerns recovering a low-rank $m \times n$ matrix from partial, noisy observation of it. This problem has been extremely well studied [32, 8, 9, 23, 10, 12, 15, 11]. However, most recovery guarantees are given in terms of Frobenius norm of the error, or mean squared error. In this work, we need reliable estimation for *each* entry, i.e., $\ell_\infty$ error bound. This is technically hard and there are only limited results [16, 13]. To make matters worse, the measurement noise in our setting can be arbitrary (not necessarily zero mean) though bounded. Thus, a new method is required and that is precisely what we do in this work. See Appendix G for more detailed discussions on why existing ME methods do not work and ours does, along with directions for future research.

## 2   Markov Decision Process, Representation of $Q$-function

**Markov Decision Process (MDP).** We consider the standard setup of infinite-horizon discounted MDP, which is described by $(\mathcal{S}, \mathcal{A}, \mathcal{P}, R, \gamma)$. $\mathcal{S}$ and $\mathcal{A}$ are the state and action spaces, respectively. $\mathcal{P}(s'|s, a)$ is the unknown transition kernel, while $R(s, a)$ determines the immediate reward received. Finally, $\gamma \in (0, 1)$ is the discounting factor. A policy $\pi(a|s)$ specifies the probability of selecting action $a \in \mathcal{A}$ at state $s \in \mathcal{S}$. The standard value function associated with a policy $\pi$ is defined as $V^\pi(s) = \mathbb{E}_\pi[\sum_{t=0}^\infty \gamma^t R(s_t, a_t) \mid s_0 = s]$. The optimal value function, denoted by $V^*$, is the value function of the reward-maximizing policy. That is, $V^*(s) = \sup_\pi V^\pi(s), \forall s \in \mathcal{S}$. Correspondingly, we define the optimal $Q$-function, denoted by $Q^*$, as $Q^*(s, a) = R(s, a) + \gamma \mathbb{E}_{s' \sim \mathcal{P}(\cdot|s,a)}[V^*(s')]$.

**MDP Regularity.** Throughout this paper, we assume the existence of a generative model (i.e., a simulator) [21]. We consider MDPs with the following properties: (1) (Compact domain) The

state space $\mathcal{S}$ and the action space $\mathcal{A}$ are compact subsets of a Euclidean space; Without loss of generality, let $\mathcal{S} = [-1, 1]^{d_1}$ and $\mathcal{A} = [-1, 1]^{d_2}$. (2) (Bounded reward) For every $(s, a) \in \mathcal{S} \times \mathcal{A}$, the reward $R(s, a)$ is bounded, i.e., $|R(s, a)| \leq R_{\max}$. (3) (Smoothness) The optimal $Q$-function, $Q^*$, is $L$-Lipschitz with respect to the 1-product metric in $\mathcal{S} \times \mathcal{A}$, i.e., $|Q^*(s_1, a_1) - Q^*(s_2, a_2)| \leq L d_{\mathcal{S} \times \mathcal{A}}\big((s_1, a_1), (s_2, a_2)\big)$ where $d_{\mathcal{S} \times \mathcal{A}}\big((s_1, a_1), (s_2, a_2)\big) = \|s_1 - s_2\|_2 + \|a_1 - a_2\|_2$.

We note that the bounded reward implies that for any policy $\pi$, $|V^\pi(s)| \leq V_{\max} \triangleq R_{\max}/(1 - \gamma)$ for all $s$. This yields $|Q^*(s, a)| \leq V_{\max}$, too. Finally, we remark that for learning MDPs with continuous state/action space under $\ell_\infty$ guarantee, some form of smoothness assumption, such as the Lipschitz continuity above, is natural and typical [51, 1, 36, 35, 18].

**Representation of $Q^*$.** With the discussion above, $Q^* : [-1, 1]^{d_1} \times [-1, 1]^{d_2} \to \mathbb{R}$ is $L$-Lipschitz and also bounded. As introduced earlier, it induces an integral kernel operator $K = K_{Q^*} : L^2([-1, 1]^{d_1}) \to L^2([-1, 1]^{d_2})$ between the spaces of square integrable functions $L^2([-1, 1]^d)$ (for $d \in \{d_1, d_2\}$) endowed with the standard inner product $\langle f, g \rangle = \int_{x \in [-1, 1]^d} f(x) g(x) dx$. Through this lens, we obtain the following representation for $Q^*$, noticing that $K$ is a Hilbert-Schmidt operator.

**Theorem 1.** *Suppose the MDP regularity conditions (1) - (3). Then there exist a nonincreasing sequence $(\sigma_i \geq \mathbb{R}_+ : i \in \mathbb{N})$ with $\sum_{i=1}^\infty \sigma_i^2 < \infty$ and orthonormal sets of functions $\{f_i \in L^2([-1, 1]^{d_1}) : i \in \mathbb{N}\}$ and $\{g_i \in L^2([-1, 1]^{d_2}) : i \in \mathbb{N}\}$ such that*

$$Q^*(s, a) = \sum_{i=1}^\infty \sigma_i f_i(s) g_i(a), \quad \forall (s, a) \in [-1, 1]^{d_1} \times [-1, 1]^{d_2}. \tag{2}$$

*As a result, for any $\delta > 0$, there exists $r^* = r^*(\delta) \in \mathbb{N}$ such that for all $r \geq r^*$, the rank-$r$ approximation error satisfies $\int_{\mathcal{S} \times \mathcal{A}} \big( \sum_{i=1}^r \sigma_i f_i(s) g_i(a) - Q^*(s, a) \big)^2 ds\, da = \sum_{i=r+1}^\infty \sigma_i^2 \leq \delta$.*

**Low Rank $Q^*$.** Theorem 1 motivates us to consider low-rank $Q^*$. For any integer $r \geq 1$, we call $Q^*$ to have rank $r$ if $\sigma_i = 0$ for all $i > r$ in (2). More generally, we say $Q^*$ has $\delta$-approximate rank $r$ if $r^*(\delta) = r$ in Theorem 1. We focus on efficient RL for $Q^*$ with exact or approximate low rank $r$. To motivate the readers, we present an example of classical MDPs that exhibits low-rank structure in $Q^*$.

**Example 1.** *The linear quadratic regulator (LQR) problem considers designing a linear controller $\pi$ for a linear dynamical system given by $s_{t+1} = As_t + Ba_t$, $a_t = \pi s_t$, by minimizing a quadratic cost (negative reward) function $R(s_t, a_t) = s_t^T E s_t + a_t^T F a_t$. Here, $s_t \in \mathbb{R}^{d_1}$ is the state of the system at time $t$, $a_t \in \mathbb{R}^{d_2}$ is the control input to the system at $t$, $A \in \mathbb{R}^{d_1 \times d_1}, B \in \mathbb{R}^{d_1 \times d_2}, \pi \in \mathbb{R}^{d_2 \times d_1}$ are matrices describing the system, and $E \in \mathbb{R}^{d_1 \times d_1}, F \in d_2 \times d_2$ are symmetric positive definite matrices. According to linear-quadratic control theory [5], the value function can be expressed as $V^\pi(s_t) = s_t^T K_\pi s_t$ where $K_\pi$ is a cost matrix for policy $\pi$; thus, the Q-function for $\pi$ is written as*

$$\begin{aligned} Q^\pi(s, a) &= R(s, a) + \gamma V^\pi\big((As + Ba)\big) \\ &= s^T (E + \gamma A^T K_\pi A) s + 2\gamma s^T A^T K_\pi B a + a^T (F + \gamma B^T K_\pi B) a. \end{aligned}$$

*Letting $A^T K_\pi B = \sum_{i=1}^r \tau_i u_i v_i^T$ be the SVD of $A^T K_\pi B$, we can see that*

$$Q^\pi(s, a) = 2\gamma \sum_{i=1}^r \tau_i (u_i^T s) \cdot (v_i^T a) + (s^T M_{\mathcal{S}} s) \cdot 1_{\mathcal{A}}(a) + 1_{\mathcal{S}}(s) \cdot (a^T M_{\mathcal{A}} a)$$

*where $M_{\mathcal{S}} = E + \gamma A^T K_\pi A$, $M_{\mathcal{A}} = F + \gamma B^T K_\pi B$ and $1_{\mathcal{S}}$ ($1_{\mathcal{A}}$) denotes a constant-1 function on $\mathcal{S}$ ($\mathcal{A}$). It is easy to observe that $1_{\mathcal{S}}$, $s^T M_{\mathcal{S}} s$, and $\{u_i^T s\}_{i=1}^r$ form an orthogonal set in $L^2(\mathcal{S})$ as long as $\mathcal{S}$ is symmetric (i.e., $\mathcal{S} = -\mathcal{S}$). Similarly, $1_{\mathcal{A}}$, $a^T M_{\mathcal{A}} a$, and $\{v_i^T a\}_{i=1}^r$ form an orthogonal set in $L^2(\mathcal{A})$ when $\mathcal{A}$ is symmetric. Thus, the rank of $Q^\pi$ is at most $\min\{d_1, d_2\} + 2$, and so is the rank of $Q^*$, which is significantly smaller than $|\mathcal{S}| \sim 2^{d_1}$ or $|\mathcal{A}| \sim 2^{d_2}$ (after quantization).*

## 3 Reinforcement Learning using Matrix Estimation

We introduce an RL algorithm using generic ME procedures as a subroutine. We require the ME method in use to satisfy Assumption 1 (see Section 4) to provide meaningful performance guarantees. However, there is no known ME procedure satisfying Assumption 1 in the literature. In Section 5, we introduce a simple ME procedure that satisfies it when $Q^*$ is exactly or approximately low-rank.

The RL algorithm iteratively improves estimation of $Q^*$. Each iteration consists of four steps: discretization, exploration, matrix estimation and generalization. We provide a narrative overview of the algorithm; the pseudo-code can be found in Appendix A.

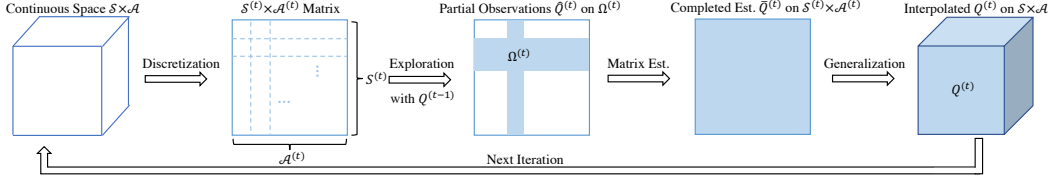

Figure 1: Iterative RL using ME: exploration uses estimation $Q^{(t-1)}$ from previous iteration.

**Step 1. Discretization.** At iteration $t$, we produce $\beta^{(t)}$-nets, $\mathcal{S}^{(t)} \subset \mathcal{S}$ and $\mathcal{A}^{(t)} \subset \mathcal{A}$, for properly chosen resolution $\beta^{(t)} \in (0,1)$ that decreases with $t$. In our setup, $|\mathcal{S}^{(t)}| = O\big((1/\beta^{(t)})^{d_1}\big)$, $|\mathcal{A}^{(t)}| = O\big((1/\beta^{(t)})^{d_2}\big)$. In total, this produces $|\mathcal{S}^{(t)}||\mathcal{A}^{(t)}|$ many $(s,a)$ pairs in the discretized set $\mathcal{S}^{(t)} \times \mathcal{A}^{(t)}$.

**Step 2. Exploration.** Using estimate $Q^{(t-1)}$ over the entire $\mathcal{S} \times \mathcal{A}$ from the previous iteration, we wish to produce an improved estimate of $Q^*$ over $\mathcal{S}^{(t)} \times \mathcal{A}^{(t)}$ through this and the next step, and then generalize it to $\mathcal{S} \times \mathcal{A}$ in Step 4. To produce an improved estimate over $\mathcal{S}^{(t)} \times \mathcal{A}^{(t)}$ in a sample-efficient manner, we first "explore" a carefully selected subset $\Omega^{(t)} \subset \mathcal{S}^{(t)} \times \mathcal{A}^{(t)}$. Specifically, for each $(s,a) \in \Omega^{(t)}$, we obtain $N^{(t)}$ samples of independent transitions using the generative model, which results in a set of sampled next states $\{s_i'\}_{i=1,\dots,N^{(t)}}$. We obtain an estimate $\hat{Q}^{(t)}(s,a)$ as

$$\hat{Q}^{(t)}(s,a) \leftarrow R(s,a) + \gamma \cdot \frac{1}{N^{(t)}} \sum_{i=1}^{N^{(t)}} V^{(t-1)}(s_i'), \quad \text{with } V^{(t-1)}(s) = \max_a Q^{(t-1)}(s,a). \quad (3)$$

**Step 3. Matrix Estimation.** Given $\hat{Q}^{(t)}(s,a), \forall (s,a) \in \Omega^{(t)}$ updated in Step 2, we wish to obtain an improved estimate of $Q^*$ for the entire $\mathcal{S}^{(t)} \times \mathcal{A}^{(t)}$. This can be viewed as a matrix estimation problem. When $Q^*$ has rank $r$ as discussed in Section 2, the sampled matrix $[Q^*(s,a) : s \in \mathcal{S}^{(t)}, a \in \mathcal{A}^{(t)}]$, induced by discretization, has rank at most $r$. Thus, we want to estimate the low-rank matrix by having access to noisy measurements for a subset of entries in $\Omega^{(t)} \subset \mathcal{S}^{(t)} \times \mathcal{A}^{(t)}$. Specifically, the noise in the measurements are not necessarily i.i.d. as they are coupled through $V^{(t-1)}$; thus, they are bounded but can be arbitrary. Ideally, we wish to obtain estimates for all entries in the matrix with the maximum error at a similar level as that in $\hat{Q}^{(t)}(s,a)$. This demands that the ME method in use is well-behaved in the $\ell_\infty$ sense, satisfying Assumption 1 as stated later. While this is absent in the literature, we shall describe ME methods fulfilling the desideratum in Sections 5.1, 5.2 and Appendix F. As a result, we obtain improved estimates $\bar{Q}^{(t)}(s,a)$ for all $(s,a) \in \mathcal{S}^{(t)} \times \mathcal{A}^{(t)}$ after the ME step.

**Step 4. Generalization.** With estimates $\bar{Q}^{(t)}(s,a)$, $(s,a) \in \mathcal{S}^{(t)} \times \mathcal{A}^{(t)}$, we generalize to $\mathcal{S} \times \mathcal{A}$ via interpolating them. This can be achieved by any supervised learning algorithm. We simply utilize the 1-nearest neighbor: for any $(s,a) \in \mathcal{S} \times \mathcal{A}$, at the end of iteration $t$ we output $Q^{(t)}(s,a) \leftarrow \bar{Q}^{(t)}(s',a')$ where $(s',a')$ is closest to $(s,a)$ in $\mathcal{S}^{(t)} \times \mathcal{A}^{(t)}$, with ties broken arbitrarily.

## 4  Main Result: Correctness, Convergence & Sample Complexity

In this section, we state the result establishing correctness, convergence and finite sample analysis of our RL algorithm. We require a specific property, stated as Assumption 1, for the Matrix Estimation (ME) method utilized in Step 3 of the algorithm. While there is no known ME method in the literature that satisfies it, we provide novel ME method with the desired property in Section 5.

**Property of Matrix Estimation.** Recall that we describe our RL algorithm with a generic matrix estimation subroutine in Step 3, without specifying what ME method is used. In fact, the success of the RL algorithm hinges on the performance of the ME method in use. For the convenience of exposition, we define $(\mathsf{C}_{\mathrm{me}}, \mathsf{c}_{\mathrm{me}})$-property of an ME method for $\mathsf{C}_{\mathrm{me}}, \mathsf{c}_{\mathrm{me}} \geq 0$, which abstracts the 'success' of the ME method and serves as a pivotal premise for the success of the entire RL algorithm.

**Assumption 1** (($C_{me}, c_{me}$)-property)**.** *Given finite $\mathcal{S}^{(t)} \subset \mathcal{S}$, $\mathcal{A}^{(t)} \subset \mathcal{A}$, it is possible to construct $\Omega^{(t)} \subseteq \mathcal{S}^{(t)} \times \mathcal{A}^{(t)}$ with $|\Omega^{(t)}| \leq C_{me}(|\mathcal{S}^{(t)}| + |\mathcal{A}^{(t)}|)$ for given constant $C_{me} \geq 1$ so that whenever the ME method in use takes $\{\hat{Q}^{(t)}(s,a)\}_{(s,a)\in\Omega^{(t)}}$ with $\max_{(s,a)\in\Omega^{(t)}} |\hat{Q}^{(t)}(s,a) - Q^*(s,a)| \leq \epsilon$ as an input and outputs $\{\bar{Q}^{(t)}(s,a)\}_{(s,a)\in\mathcal{S}^{(t)}\times\mathcal{A}^{(t)}}$, the following inequality holds:*

$$\max_{(s,a)\in\mathcal{S}^{(t)}\times\mathcal{A}^{(t)}} |\bar{Q}^{(t)}(s,a) - Q^*(s,a)| \leq c_{me}\epsilon.$$

We assume access to an ME method that satisfies ($C_{me}, c_{me}$)-property. Assumption 1 ensures the $\ell_\infty$ error remains under control (to be precise, $c_{me}$-Lipschitz with respect to $\ell_\infty/\ell_\infty$) during the ME step, while it is stated in the language of RL for later uses. Note that Assumption 1 does not explicitly require any structure on $Q^*$, but we will require $Q^*$ to be low-rank or approximately low-rank to produce an ME method satisfying it, as will be discussed in Section 5.

**Correctness, Convergence & Sample Complexity of RL Algorithm.** Now, we state the desired properties of the RL algorithm introduced in Section 3. To that end, let the algorithm start with initialization $Q^{(0)}(s,a) = 0$, $\forall (s,a) \in \mathcal{S} \times \mathcal{A}$ and hence $V^{(0)}(s) = 0$, $\forall s \in \mathcal{S}$. That is, $|Q^{(0)}(s,a) - Q^*(s,a)| \leq V_{\max}$, $\forall (s,a) \in \mathcal{S} \times \mathcal{A}$. For the sake of notational brevity, we let $d_1 = d_2 = d$ in the sequel. We remark that our theorems apply equally by simply replacing $d$ with $\max\{d_1, d_2\}$.

**Theorem 2.** *Consider the RL algorithm described in Section 3 with ME satisfying Assumption 1. Given $\delta \in (0,1)$, there exists algorithmic choice of $\beta^{(t)}, \Omega^{(t)}, N^{(t)}$ for $1 \leq t \leq T$, so that*

$$\mathbb{P}\left( \sup_{(s,a)\in\mathcal{S}\times\mathcal{A}} |Q^{(t)}(s,a) - Q^*(s,a)| \leq (2\gamma c_{me})^t V_{\max}, \quad \forall 1 \leq t \leq T \right) \geq 1 - \delta. \qquad (4)$$

*Further, let $\gamma < \frac{1}{2c_{me}}$. Then, with $T = \Theta(\log\frac{1}{\epsilon})$ and $\widetilde{O}(\frac{1}{\epsilon^{d+2}} \cdot \log\frac{1}{\delta})$ number of samples, we have*

$$\mathbb{P}\left( \sup_{(s,a)\in\mathcal{S}\times\mathcal{A}} |Q^{(T)}(s,a) - Q^*(s,a)| \leq \epsilon \right) \geq 1 - \delta. \qquad (5)$$

In the proof of Theorem 2 presented in Appendix C, we choose parameters $\beta^{(t)} = \frac{V_{\max}}{8L}(2\gamma c_{me})^t$, $|\Omega^{(t)}| = C_{me}(|\mathcal{S}^{(t)}| + |\mathcal{A}^{(t)}|)$ and $N^{(t)} = \frac{8}{(2\gamma c_{me})^{2(t-1)}} \log\left(\frac{2|\Omega^{(t)}|T}{\delta}\right)$ for $1 \leq t \leq T$. While this choice establishes the claims in Theorem 2, it is possible to achieve $\sup_{(s,a)\in\mathcal{S}\times\mathcal{A}} |Q^{(t)}(s,a) - Q^*(s,a)| \leq \alpha^t V_{\max}$ for any $\alpha > \gamma c_{me}$ by making a more sophisticated choice. Subsequently, the conclusion for sample complexity, (5), can be extended for any $\gamma < \frac{1}{c_{me}}$. Thus, the constant $c_{me}$ in Assumption 1 determines the range of MDPs for which such gains can be achieved. In our analysis of the proposed ME method, $c_{me} \geq 1$ and indeed, we can achieve $c_{me} = 1$ by trivially selecting $\Omega^{(t)} = \mathcal{S}^{(t)} \times \mathcal{A}^{(t)}$, which however, does not lead to any gain in efficiency. The key challenge is to find the right balance between small $c_{me}$ with small $|\Omega^{(t)}|$ or $C_{me}$. We address this next.

## 5 Matrix Estimation Satisfying Assumption 1

We introduce matrix estimation method satisfying Assumption 1 which is required for the success of our RL algorithm as in Theorem 2. For ease of illustration, we start with describing it for the rank-1 setting, then generalize it for $Q^*$ with generic rank $r \geq 1$ and finally for approximate rank-$r$ setting.

### 5.1 Matrix Estimation for $Q^*$ with Rank 1: A Warm-Up

Consider $Q^*$ with rank 1. That is, there exist $f : \mathcal{S} \to \mathbb{R}$ and $g : \mathcal{A} \to \mathbb{R}$ so that $Q^*(s,a) = f(s)g(a)$ for all $(s,a) \in \mathcal{S} \times \mathcal{A}$. For the ease of exposition, we assume $R(s,a) \in [R_{\min}, R_{\max}]$ with $R_{\min} > 0$ for all $(s,a) \in \mathcal{S} \times \mathcal{A}$ in this warm-up only. Subsequently, $Q^*(s,a) \geq V_{\min} \triangleq \frac{R_{\min}}{1-\gamma}$, $\forall (s,a)$.

**Matrix Estimation Algorithm.** For $t \geq 1$, consider a discretization of spaces, $\mathcal{S}^{(t)} \subset \mathcal{S}$, $\mathcal{A}^{(t)} \subset \mathcal{A}$. Let $Q^*(\mathcal{S}^{(t)}, \mathcal{A}^{(t)})$ be the $|\mathcal{S}^{(t)}| \times |\mathcal{A}^{(t)}|$ matrix induced by restricting $Q^*$ to $\mathcal{S}^{(t)} \times \mathcal{A}^{(t)}$. Since $Q^*$ is rank 1, it follows that $Q^*(\mathcal{S}^{(t)}, \mathcal{A}^{(t)}) = FG^T$ where $F = [f(s) : s \in \mathcal{S}^{(t)}] \in \mathbb{R}^{|\mathcal{S}^{(t)}|\times 1}$ and $G = [g(a) : a \in \mathcal{A}^{(t)}] \in \mathbb{R}^{|\mathcal{A}^{(t)}|\times 1}$. Therefore, we can estimate $Q^*(\mathcal{S}^{(t)}, \mathcal{A}^{(t)})$ by estimating $F, G$.

Now we describe the selection of $\Omega^{(t)}$ such that $|\Omega^{(t)}| = |\mathcal{S}^{(t)}| + |\mathcal{A}^{(t)}| - 1$. To that end, we first choose *anchor* elements $s^\sharp \in \mathcal{S}^{(t)}$, $a^\sharp \in \mathcal{A}^{(t)}$ and let $\Omega^{(t)} = \{(s, a) \in \mathcal{S}^{(t)} \times \mathcal{A}^{(t)} : s = s^\sharp \text{ or } a = a^\sharp\}$. With access to $\{\hat{Q}^{(t)}(s, a) : (s, a) \in \Omega^{(t)}\}$, our ME method produces estimates for all $(s, a) \in \mathcal{S}^{(t)} \times \mathcal{A}^{(t)}$ as $\bar{Q}^{(t)}(s, a) = \frac{\hat{Q}^{(t)}(s, a^\sharp)\hat{Q}^{(t)}(s^\sharp, a)}{\hat{Q}^{(t)}(s^\sharp, a^\sharp)}$.

**Satisfaction of Assumption 1.** For the algorithm described above, we state the following proposition which verifies that Assumption 1 is satisfied with $\mathsf{C}_{\mathsf{me}} = 1$ and $\mathsf{c}_{\mathsf{me}} = 7\frac{R_{\max}}{R_{\min}}$.

**Proposition 3.** *For $\epsilon \leq \frac{1}{2}V_{\min}$, suppose that $\max_{(s,a) \in \Omega^{(t)}} \left|\hat{Q}^{(t)}(s, a) - Q^*(s, a)\right| \leq \epsilon$. Then the estimate produced by the above ME algorithm satisfies*

$$\max_{(s,a) \in \mathcal{S}^{(t)} \times \mathcal{A}^{(t)}} \left|\bar{Q}^{(t)}(s, a) - Q^*(s, a)\right| \leq 7\frac{R_{\max}}{R_{\min}}\epsilon.$$

Proposition 3 implies that when $Q^*$ has rank 1, our simple ME method satisfies $\left(1, 7\frac{R_{\max}}{R_{\min}}\right)$-property for $\epsilon \leq \frac{1}{2}V_{\min}$. We remark that for any $c \in (0, 1)$, the method fulfills $\left(1, \mathsf{c}_{\mathsf{me}}\right)$-property with $\mathsf{c}_{\mathsf{me}} = \frac{3+c}{1-c}\frac{R_{\max}}{R_{\min}}$ for all $\epsilon \leq cV_{\min}$. Replacing Assumption 1 in Theorem 2 with Proposition 3, we get convergence & sample complexity guarantees for the rank-1 setup (Theorem 9 in Appendix D).

## 5.2 Matrix Estimation for $Q^*$ with Rank $r$

Based on the intuition developed in Section 5.1, we consider a more general rank-$r$ setup. For notational convenience, given $Q : \mathcal{S} \times \mathcal{A} \to \mathbb{R}$ and $\mathcal{S}' \subset \mathcal{S}, \mathcal{A}' \subset \mathcal{A}$, we let $Q(\mathcal{S}', \mathcal{A}')$ denote the $|\mathcal{S}'| \times |\mathcal{A}'|$ matrix $[Q(s, a) : (s, a) \in \mathcal{S}' \times \mathcal{A}']$, whose entries are indexed by $(s, a) \in \mathcal{S}' \times \mathcal{A}'$.

The central idea is the same as before: although $Q^*(\mathcal{S}^{(t)}, \mathcal{A}^{(t)}) \in \mathbb{R}^{m \times n}$ is an array of $mn$ numbers, it has only $r(m + n - r)$ degrees of freedom, when $\text{rank}(Q^*(\mathcal{S}^{(t)}, \mathcal{A}^{(t)})) = r \leq \min\{m, n\}$; as a result, one can restore $Q^*(\mathcal{S}^{(t)}, \mathcal{A}^{(t)})$ by exploring only $r$ entire rows and columns. There is, however, a small caveat that the $r$ rows and $r$ columns should be carefully chosen so that they are not degenerate, i.e., the $r$ rows span the entire row space of $Q^*(\mathcal{S}^{(t)}, \mathcal{A}^{(t)})$ (the $r$ columns span the column space of $Q^*(\mathcal{S}^{(t)}, \mathcal{A}^{(t)})$, resp.). To this end, we define the notion of anchor states and actions.

**Definition 4.** *(Anchor states and actions) A set of states $\mathcal{S}^\sharp = \{s_i^\sharp\}_{i=1}^{R_s} \subset \mathcal{S}$ and actions $\mathcal{A}^\sharp = \{a_i^\sharp\}_{i=1}^{R_a} \subset \mathcal{A}$ for some $R_s, R_a$ are called anchor states and actions for $Q^*$ if $\text{rank } Q^*(\mathcal{S}^\sharp, \mathcal{A}^\sharp) = r$.*

That is, there are $r$ states in the set $\mathcal{S}^\sharp$ such that $Q^*(s, \mathcal{A}^\sharp), s \in \mathcal{S}^\sharp$ are linearly independent. In other words, $\mathcal{S}^\sharp$ contains states with sufficiently diverse performance on actions $\mathcal{A}^\sharp$. Likewise, a similar interpretation holds for $\mathcal{A}^\sharp$ if we look at the columns of $Q^*(\mathcal{S}^\sharp, \mathcal{A}^\sharp)$.

Indeed, $\mathcal{S}^\sharp$ and $\mathcal{A}^\sharp$ will be used to construct our exploration sets and we want them to have small size. Finding only a few diverse states and actions is arguably easy in practice — in fact, for stochastic control tasks experimented in Section 6, we simply pick a few states and actions that are far from each other in their respective metric. We remark that assuming some "anchor" elements (i.e., elements having some special, relevant properties) is common in feature-based reinforcement learning [48, 17] or matrix factorization such as topic modeling [2].

**Matrix Estimation Algorithm.** We select anchor states $\mathcal{S}^\sharp \subset \mathcal{S}$, anchor actions $\mathcal{A}^\sharp \subset \mathcal{A}$ and fix them throughout all iterations $1 \leq t \leq T$. As before, we select appropriate $\beta^{(t)}$-nets $\mathcal{S}^{(t)}$ and $\mathcal{A}^{(t)}$ and augment them with the anchor states and actions: $\bar{\mathcal{S}}^{(t)} \leftarrow \mathcal{S}^{(t)} \cup \mathcal{S}^\sharp$ and $\bar{\mathcal{A}}^{(t)} \leftarrow \mathcal{A}^{(t)} \cup \mathcal{A}^\sharp$. For iteration $1 \leq t \leq T$, we let $\Omega^{(t)} = \{(s, a) \in \bar{\mathcal{S}}^{(t)} \times \bar{\mathcal{A}}^{(t)} : s \in \mathcal{S}^\sharp \text{ or } a \in \mathcal{A}^\sharp\}$ be the exploration set.

Given $\hat{Q}^{(t)}(s, a)$ for $(s, a) \in \Omega^{(t)}$, our ME method produces estimates for all $(s, a) \in \mathcal{S}^{(t)} \times \mathcal{A}^{(t)}$ as

$$\bar{Q}^{(t)}(s, a) = \hat{Q}^{(t)}(s, \mathcal{A}^\sharp)\left[\hat{Q}^{(t)}(\mathcal{S}^\sharp, \mathcal{A}^\sharp)\right]^\dagger \hat{Q}^{(t)}(\mathcal{S}^\sharp, a) \tag{6}$$

where $X^\dagger$ denotes the Moore-Penrose pseudoinverse of the matrix $X$. With the choice of $R_s = R_a = r$ (or a constant multiple of $r$), the size of $\Omega^{(t)}$ is at most $r(|\bar{\mathcal{S}}^{(t)}| + |\bar{\mathcal{A}}^{(t)}| - r) \ll |\bar{\mathcal{S}}^{(t)}||\bar{\mathcal{A}}^{(t)}|$.

**Satisfaction of Assumption 1.** For given matrix $X \in \mathbb{R}^{m \times n}$, we denote by $\sigma_i(X)$ its $i$-th largest singular value, i.e., $\sigma_1(X) \geq \sigma_2(X) \geq \cdots \geq \sigma_{\min(m,n)}(X) \geq 0$. We state the following guarantee, which verifies that the matrix estimation algorithm described above satisfies Assumption 1.

**Proposition 5** (Simplified version of Proposition 10). *Let $\Omega^{(t)}$ and $\bar{Q}^{(t)}$ as described above and let $|\mathcal{S}^\sharp| = |\mathcal{A}^\sharp| = r$. For any $\epsilon \leq \frac{1}{2r}\sigma_r\big(Q^*(\mathcal{S}^\sharp, \mathcal{A}^\sharp)\big)$, if $\max_{(s,a)\in\Omega^{(t)}} \big|\hat{Q}^{(t)}(s,a) - Q^*(s,a)\big| \leq \epsilon$, then*

$$\max_{(s,a)\in\mathcal{S}^{(t)}\times\mathcal{A}^{(t)}} \big|\bar{Q}^{(t)}(s,a) - Q^*(s,a)\big| \leq c(r;\mathcal{S}^\sharp,\mathcal{A}^\sharp)\epsilon$$

*where $c(r;\mathcal{S}^\sharp,\mathcal{A}^\sharp) = \Big(6\sqrt{2}\big(\frac{r}{\sigma_r(Q^*(\mathcal{S}^\sharp,\mathcal{A}^\sharp))}\big) + 2(1+\sqrt{5})\big(\frac{r}{\sigma_r(Q^*(\mathcal{S}^\sharp,\mathcal{A}^\sharp))}\big)^2\Big)V_{\max}$.*

Proposition 5 implies when $Q^*$ has rank $r$, our ME method satisfies $\big(r, c(r;\mathcal{S}^\sharp,\mathcal{A}^\sharp)\big)$-property for $\epsilon \leq \frac{1}{2r}\sigma_r\big(Q^*(\mathcal{S}^\sharp,\mathcal{A}^\sharp)\big)$. Hence, we obtain Theorem 11 (Appendix E.2) as a corollary of Theorem 2. That is, we obtain the desired convergence and sample complexity $\tilde{O}(\frac{1}{\epsilon^{d+2}} \cdot \log\frac{1}{\delta})$ to achieve $\epsilon$ error with the output $Q^{(T)}$. Our algorithm and analysis also apply to low-rank $Q^*$ over discrete spaces. We summarize results for (1) continuous $\mathcal{S}$ and finite $\mathcal{A}$; (2) finite $\mathcal{S}$ and $\mathcal{A}$ in Appendix E.3.

### 5.3 Matrix Estimation for $Q^*$ with Approximate Rank $r$ (Appendix F)

In practice, it may not be feasible to hope for exact low-rank structure. Hence, it is desirable to seek methods that are reasonably robust to approximation error. We show that our ME method has such an appealing property. Here, we provide a concise summary with full details deferred to Appendix F.

Given $r > 0$ as a parameter, let $Q_r^*$ denote the best rank-$r$ approximation of $Q^*$ (cf. Theorem 1). Denote by $\zeta_r \triangleq \sup_{(s,a)\in\mathcal{S}\times\mathcal{A}} |Q_r^*(s,a) - Q^*(s,a)|$ the model bias due to approximation. We generalize the notion of anchor states and actions to $r$-anchor states and actions: we call $\mathcal{S}^\sharp$ and $\mathcal{A}^\sharp$ to be $r$-anchor states and actions if rank $Q_r^*(\mathcal{S}^\sharp, \mathcal{A}^\sharp) = r$. Let $c_{\mathrm{appx}}(r;\mathcal{S}^\sharp,\mathcal{A}^\sharp)$ denote a constant defined in the same way as $c(r;\mathcal{S}^\sharp,\mathcal{A}^\sharp)$ in Proposition 5, with $\sigma_r\big(Q^*(\mathcal{S}^\sharp,\mathcal{A}^\sharp)\big)$ replaced by $\sigma_r(Q_r^*(\mathcal{S}^\sharp,\mathcal{A}^\sharp))$. We apply the same ME algorithm described in Section 5.2 with $\mathcal{S}^\sharp, \mathcal{A}^\sharp$ being $r$-anchor states/actions. Theoretically, we guarantee that when the model bias $\zeta_r$ is sufficiently small, we obtain convergence and sample complexity results similar to the exact rank-$r$ setting with an additive error, $\frac{1+\gamma c_{\mathrm{appx}}(r;\mathcal{S}^\sharp,\mathcal{A}^\sharp)}{1-\gamma c_{\mathrm{appx}}(r;\mathcal{S}^\sharp,\mathcal{A}^\sharp)}\zeta_r$, induced by the approximation bias.

**Theorem 6** (Informal Statement of Theorem 16). *Consider the approximate rank-$r$ setting. Suppose $\gamma \leq \frac{1}{2c_{appx}(r;\mathcal{S}^\sharp,\mathcal{A}^\sharp)}$ and let $\delta \in (0,1)$. Under certain regularity conditions on $Q_r^*$ and $\zeta_r$, we have: for any $\epsilon > 0$, with $T = \Theta(\log\frac{1}{\epsilon})$ and $\tilde{O}(\frac{1}{\epsilon^{d+2}} \cdot \log\frac{1}{\delta})$ number of samples, we obtain estimate such that*

$$\mathbb{P}\left(\sup_{(s,a)\in\mathcal{S}\times\mathcal{A}} \big|Q^{(T)}(s,a) - Q^*(s,a)\big| \leq \epsilon + \frac{1+\gamma c_{appx}(r;\mathcal{S}^\sharp,\mathcal{A}^\sharp)}{1-\gamma c_{appx}(r;\mathcal{S}^\sharp,\mathcal{A}^\sharp)}\zeta_r\right) \geq 1 - \delta.$$

## 6 Empirical Evaluation

Besides theory, we empirically validate the effectiveness of our method on 5 continuous control tasks. The detailed setup can be found in Appendix H. In short, we first discretize the spaces into very fine grid and run standard value iteration to obtain a proxy of $Q^*$. The proxy has a very small approximate rank in all tasks; we hence use $r = 10$ for our experiments. As mentioned, we simply select $r$ states and $r$ actions that are far from each other in their respective spaces as our anchor states and actions. For example, if the space is 2-dimensional, we uniformly divide it into $r$ squares and sample one from each square. Because of unavoidable discretization error, we also provide results on mean error, which might be a more reasonable measure in practice. While our proof requires small $\gamma$, we find the method to be generally applicable with large $\gamma$ in real tasks. Therefore, we use $\gamma = 0.9$ in all the tasks. Additional results on this aspect as well as results on all 5 tasks are provided in Appendix I.

**Improved Sample Complexity with ME.** First, we confirm that the sample complexity of our algorithm improves with the use of ME. Our baseline algorithm refers to the same algorithm described in Section 3, but without the ME step (Step 3); i.e., we explore and update all $(s,a) \in \mathcal{S}^{(t)} \times \mathcal{A}^{(t)}$, which is equivalent to performing a simulated value iteration on the entire discretized set (a.k.a. the synchronous model for Q learning). We illustrate the sample complexity for achieving different levels of $\ell_\infty$ error (Figure 2(a)) and mean error (Figure 2(b)). It is clear from the plots that our algorithm uses significantly less samples to achieve error at a similar level to the baseline. This evidences that exploiting structure leads to improved efficiency. The same conclusion holds for the other tasks.

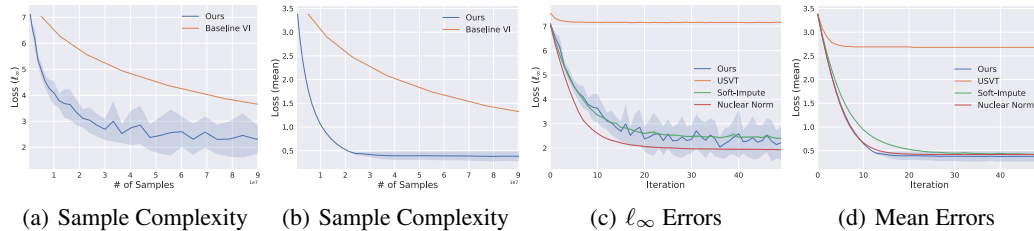

|                        |                        |                   |              |
|:----------------------:|:----------------------:|:-----------------:|:------------:|
| (a) Sample Complexity  | (b) Sample Complexity  | (c) $\ell_\infty$ Errors | (d) Mean Errors |

Figure 2: Empirical results on the Inverted Pendulum control task. In (a) and (b), we show the improved sample complexity for achieving different levels of $\ell_\infty$ error and mean error, respectively. In (c) and (d), we compare the $\ell_\infty$ error and the mean error for various ME methods. Results are averaged across 5 runs for each method.

**Error Guarantees.** Next, we compare our ME method with other popular ME methods to validate its performance. While theoretically insufficient for RL applications, some established ME methods [8, 10, 27] work well in practice. We compare the methods by feeding the same number of samples of size $O(\max\{\mathcal{S}^{(t)}, \mathcal{A}^{(t)}\})$. As in Figure 2(c) & 2(d), our method displays a competitive performance, both in $\ell_\infty$ & mean errors. Also, we note that our simple, but powerful method is computationally much more efficient, compared to other methods based on optimization, etc. It can be $40\times$ faster than nuclear norm minimization, cf. Table 4 in Appendix I.6. Overall, these results emphasize the practical value of our method beyond its theoretical soundness. Lastly, we remark other ME methods also show promise in our experiments; it is certainly a valuable open question to harmonize the established ME methods with low-rank RL.

**Resulting Policy.** As a final proof of concept, we observe that the eventual performance of the policy obtained from the output $Q^{(T)}$ is very close to the policy obtained from $Q^*$ (cf. plots in Appendix I). We summarize the results for standard performance metrics used in Table 3 . Obviously, our efficient method exhibits very competitive performance.

Table 3:  Performance metric for different stochastic control tasks using different ME methods. A.D. stands for angular deviation, T.G. stands for time-to-goal; for both metrics, the smaller the better.

| Method | Optimal | USVT [10] | Soft-Impute [27] | Nuclear Norm [8] | Ours |
|---|---|---|---|---|---|
| Inverted Pendulum (A.D.) | $1.6 \pm .0$ | $22.5 \pm 2.5$ | $5.3 \pm .6$ | $3.1 \pm .3$ | $3.4 \pm .7$ |
| Mountain Car (T.G.) | $75.0 \pm .3$ | $358.8 \pm 5.0$ | $168.4 \pm 8.1$ | $92.4 \pm 2.8$ | $91.8 \pm 7.2$ |
| Double Integrator (T.G.) | $199.5 \pm .1$ | $200.0 \pm .4$ | $199.9 \pm .3$ | $199.6 \pm .2$ | $199.7 \pm .4$ |
| Cart-Pole (A.D.) | $10.1 \pm .0$ | $19.2 \pm 1.0$ | $10.4 \pm .1$ | $10.2 \pm .1$ | $10.2 \pm .2$ |
| Acrobot (A.D.) | $2.4 \pm .0$ | $28.8 \pm 4.3$ | $9.1 \pm 1.2$ | $5.1 \pm .8$ | $6.2 \pm 1.0$ |

# 7    Conclusion

We provide an efficient RL framework for continuous state and action spaces via proposing a new low-rank perspective. With a novel ME method in the RL context, we demonstrate that our low-rank approach is both theoretically and practically appealing in designing sample efficient methods.

We remark that there remain several interesting open questions. First of all, we believe it is possible to refine the error analysis in this work to achieve a stronger theoretical guarantee. For example, the condition $\gamma < \frac{1}{2c_{me}}$ for convergence in Theorem 2 is probably an artifact of our decoupled analysis and is possibly removable. Perhaps, devising better ME methods for the purpose of RL can be a solution to lift the restriction on the range of $\gamma$, which is an interesting problem on its own. Also, we conjecture that our "sample and pseudo-explore (via ME)" scheme is more broadly applicable beyond the generative setup considered in this work, e.g., to the online setup. The most prominent challenge anticipated in online setup is that we are no longer able to sample "any" state-action pair freely and adaptively; the sampling needs to respect the exploration policy. This difficulty may be overcome with a more refined ME method, with ad-hoc techniques.

Overall, we believe this work can serve as a starting point for fruitful future research along the promising direction of low-rank RL.

## Broader Impact

As reinforcement learning becomes increasingly popular in practice and the problem dimension grows, there is a soaring demand for data-efficient learning algorithms. Through the lens of low-rank representation of so-called $Q$-function, this work proposes a theoretical framework to devise efficient RL algorithms. The resulting "low-rank" algorithm, which utilizes a novel matrix estimation method, offers both strong theoretical guarantees and appealing empirical performance.

In particular, the novel "low-rank" perspective about RL provides an effective tool to tackle RL problems with both state and action spaces continuous, which have received much less attention despite their practical significance. We believe that this work serves as an important step towards provable efficient RL for continuous problems. The theoretical insights in this work can motivate further research in both efficient RL and ME, while the empirical results should be beneficial more broadly for practitioners working in continuous controls.

## Acknowledgments and Disclosure of Funding

This work was supported in parts by NSF projects CMMI-1462158, CMMI-1634259, CNS-1523546, TRIPODS Phase 1; a joint project with KAIST (South Korea); and a project funded by KACST. Dogyoon Song was also partially supported by Samsung Scholarship and MIT Hewlett Packard Fellowship, and Zhi Xu was supported in part by the Siemens FutureMakers Fellowship.

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
