[Supplementary Material]

# Appendices

## A   Reinforcement Learning using Matrix Estimation: The Pseudo Code

Below is a pseudo-code of the generic RL method presented in Section 3.

---

**Algorithm 1** Main Algorithm: Low-rank Reinforcement Learning

---

$\quad$ **Input:** $\quad \mathcal{S}, \mathcal{A}, \gamma, Q^{(0)}, T, \{\beta^{(t)}\}_{t=1,\ldots,T}, \{N^{(t)}\}_{t=1,\ldots,T}$

$\quad$ **Output:** $\quad Q^{(T)}$, the $Q$-value oracle after $T$ iterations

1: **Initialization:** For all $s \in \mathcal{S}$, initialize the value oracle $Q^{(0)}(s)$.
2: **for** $t = 1, 2, \ldots, T$ **do**
3: $\quad$ /* Step 1: Discretization of $\mathcal{S}$ and $\mathcal{A}$ */
4: $\quad$ Discretize $\mathcal{S}$ and $\mathcal{A}$ so that $\mathcal{S}^{(t)}$ is a $\beta^{(t)}$-net of $\mathcal{S}$ and $\mathcal{A}^{(t)}$ is a $\beta^{(t)}$-net of $\mathcal{A}$.
5: $\quad$ /* Step 2: Exploration of a few $(s, a)$ pairs */
6: $\quad$ Select a subset of $(s, a)$ pairs, $\Omega^{(t)} \subseteq \mathcal{S}^{(t)} \times \mathcal{A}^{(t)}$.
7: $\quad$ **for** $(s, a) \in \Omega^{(t)}$ **do**
8: $\quad\quad$ Estimate $Q^*(s, a)$ via simple lookahead based on the current value oracle $V^{(t-1)}$, i.e., query the generative model to sample $N^{(t)}$ independent transitions from $(s, a)$ and obtain an estimate $\hat{Q}^{(t)}(s, a)$ with the sampled next states $\{s'_i\}_{i=1,\ldots,N^{(t)}}$:

$$\hat{Q}^{(t)}(s, a) \leftarrow R(s, a) + \gamma \cdot \frac{1}{N^{(t)}} \sum_{i=1}^{N^{(t)}} V^{(t-1)}(s'_i). \tag{7}$$

9: $\quad$ **end for**
10: $\quad$ /* Step 3: Matrix completion to obtain $\bar{Q}$ from $\hat{Q}$ */
11: $\quad$ Estimate $\bar{Q}(s, a)$ for $(s, a) \in \mathcal{S}^{(t)} \times \mathcal{A}^{(t)}$ from the data $\{\hat{Q}^{(t)}(s, a)\}_{(s,a)\in\Omega^{(t)}}$, utilizing the low-rank structure of $\bar{Q}(s, a)$, viz.,

$$\bar{Q}^{(t)} \leftarrow \text{Matrix Estimation}\big(\hat{Q}^{(t)}; \Omega^{(t)}\big).$$

12: $\quad$ /* Step 4: Generalization via interpolating $\bar{Q}$ */
13: $\quad$ Update the oracles $Q^{(t)}$ and $V^{(t)}$ by calling a subroutine that interpolates $\bar{Q}^{(t)}$ through non-parametric regression methods:

$$Q^{(t)} \leftarrow \text{Interpolation}\big(\bar{Q}^{(t)}; \mathcal{S}^{(t)}, \mathcal{A}^{(t)}\big),$$

$\quad$ and subsequently, $V^{(t)}(s) \leftarrow \max_{a \in \mathcal{A}} Q^{(t)}(s, a)$, for all $s \in \mathcal{S}$.
14: **end for**

---

## B   Proof of Theorem 1

The proof of Theorem 1 follows from the classical results in functional analysis. Interested reader may find lecture notes [6] and a classical textbook on the topic [14] as excellent references. In this section, we present and prove a more general version of Theorem 1 that is applicable to any compact metric spaces equipped with finite measures.

Let $\mathcal{S}$ and $\mathcal{A}$ be compact metric spaces, equipped with finite measures $\mu$, $\nu$, respectively. We consider the space of square integrable functions

$$L^2(\mathcal{S}, \mu) = \left\{ f : \mathcal{S} \to \mathbb{R} \text{ such that } \|f\|_{L^2(\mathcal{S}, \mu)} \equiv \left( \int_{\mathcal{S}} |f(s)|^2 d\mu(s) \right)^{\frac{1}{2}} < \infty \right\}$$

and $L^2(\mathcal{A}, \nu)$ defined similarly. $L^2(\mathcal{S}, \mu)$ and $L^2(\mathcal{A}, \nu)$ are known to be Hilbert spaces and in particular, they are separable because $\mathcal{S}$ and $\mathcal{A}$ are compact metric spaces. Therefore, they have countable bases.

Recall that given any vector space $V$ over $\mathbb{R}$, its dual space $V^*$ is defined as the set of all linear maps $\phi : V \to \mathbb{R}$. It is known that the dual of $L^2(\mathcal{S}, \mu)$ is isometrically isomorphic to $L^2(\mathcal{S}, \mu)$, e.g., by the isomorphism $f \mapsto f^*$ where $f^*(f') = \langle f', f \rangle = \int_{\mathcal{S}} f(s)f'(s)d\mu(s)$ (Appendix B, [14]).

Given two Hilbert spaces, $\mathcal{H}_1, \mathcal{H}_2$, we let $\mathcal{H}_1 \otimes \mathcal{H}_2$ denote the tensor product of the two Hilbert spaces. The inner product in $\mathcal{H}_1 \otimes \mathcal{H}_2$ is defined on the basis elements so that $\langle \phi_1 \otimes \phi_2, \psi_1 \otimes$

$\psi_2\rangle_{\mathcal{H}_1 \otimes \mathcal{H}_2} = \langle \phi_1, \psi_1 \rangle_{\mathcal{H}_1} \langle \phi_2, \psi_2 \rangle_{\mathcal{H}_2}$ for all $\phi_1, \psi_1 \in \mathcal{H}_1$ and $\phi_2, \psi_2 \in \mathcal{H}_2$. Also, for every element $\phi_1 \otimes \phi_2 \in \mathcal{H}_1 \otimes \mathcal{H}_2$, one can associate the rank-1 operator from $\mathcal{H}_1^* \to \mathcal{H}_2$ that maps a given $x^* \in \mathcal{H}_1^*$ to $x^*(\phi_1)\phi_2$.

Our main theorem in this section is the following spectral theorem (singular value theorem) for $Q^*$. It is indeed a classical result from operator theory on Hilbert spaces. However, most results in existing literature cover the theory for self-adjoint operators and symmetric kernels. Although it is already implied by the classical results in a similar manner as eigenvalue decomposition extends to singular value decomposition, here we state our theorem and its proof for readers' convenience and future references.

**Theorem 7.** *Let $(\mathcal{S}, d_\mathcal{S}, \mu)$ and $(\mathcal{A}, d_\mathcal{A}, \nu)$ be compact metric spaces equipped with finite measures. Let $Q^* \in L^2(\mathcal{S} \times \mathcal{A}, \mu \times \nu)$. If $Q^*$ is $L$-Lipschitz with respect to the product metric, then there exist a nonincreasing sequence $(\sigma_i \geq \mathbb{R}_+ : i \in \mathbb{N})$ with $\sum_{i=1}^\infty \sigma_i^2 < \infty$ and orthonormal bases $\{f_i \in L^2(\mathcal{S}, \mu) : i \in \mathbb{N}\}$ and $\{g_i \in L^2(\mathcal{A}, \nu) : i \in \mathbb{N}\}$ such that*

$$Q^* = \sum_{i=1}^\infty \sigma_i f_i \otimes g_i. \tag{8}$$

*Subsequently, for any $\delta > 0$, there exists $r^*(\delta) \in \mathbb{N}$ such that $\left\| \sum_{i=1}^r \sigma_i f_i \otimes g_i - Q^* \right\|_{L^2(\mathcal{S} \times \mathcal{A}, \mu \times \nu)}^2 \leq \delta$) for all $r \geq r^*(\delta)$.*

Note that we obtain the equality (8) in the $L^2$ sense. However, since $Q^*$ is assumed Lipschitz continuous on a compact domain, this actually gives us a pointwise equality, i.e., $Q^*(s, a) = \sum_{i=1}^\infty \sigma_i f_i(s) g_i(a)$ for all $(s, a) \in \mathcal{S} \times \mathcal{A}$.

*Proof.* We define an integral kernel operator $K = K_{Q^*} : L^2(\mathcal{S}, \mu) \to L^2(\mathcal{A}, \nu)$ induced by the kernel $Q^* \in L^2(\mathcal{S} \times \mathcal{A}, \mu \times \nu)$ so that

$$Kf(\cdot) = \int_\mathcal{S} Q^*(s, \cdot)f(s)d\mu(s).$$

Observe that $Q^*$ is a continuous function defined on a compact domain and hence bounded, viz., there exists $V_{\max} < \infty$ such that $|Q^*(s, a)| \leq V_{\max}$ for all $(s, a) \in \mathcal{S} \times \mathcal{A}$.

We present our proof in four parts. First, we verify that $K$ is a compact operator from $L^2(\mathcal{S}, \mu)$ to $L^2(\mathcal{A}, \nu)$. Next, we argue $K$ admits a generalized singular value decomposition with square summable singular values, based on the spectral theory of compact operators. Then we transfer the results for $K \in L^2(\mathcal{S}, \mu)^* \otimes L^2(\mathcal{A}, \nu)$ to argue the spectral decomposition of $Q^* \in L^2(\mathcal{S}, \mu) \otimes L^2(\mathcal{A}, \nu)$. Lastly, we conclude the proof by discussing rank-$r$ approximation of $Q^*$.

1. $K$ is a compact operator from $L^2(\mathcal{S}, \mu)$ to $L^2(\mathcal{A}, \nu)$.

   First, we argue that $K$ is a bounded linear operator with $\|K\| \leq V_{\max}^2 \mu(\mathcal{S})\nu(\mathcal{A})$. Recall that $Q^* : \mathcal{S} \times \mathcal{A} \to \mathbb{R}$ is Lipschitz continuous on a compact domain, hence, bounded, i.e., there exists $V_{\max} < \infty$ such that $|Q^*(s, a)| \leq V_{\max}$ for all $(s, a) \in \mathcal{S} \times \mathcal{A}$. For any $f \in L^2(\mathcal{S}, \mu)$,

   $$\begin{aligned}
   \|Kf\|_{L^2(\mathcal{A}, \nu)}^2 &= \int_\mathcal{A} Kf(a)^2 d\nu(a) \\
   &= \int_\mathcal{A} \left( \int_\mathcal{S} Q^*(s, a)f(s)d\mu(s) \right)^2 d\nu(a) \\
   &\leq \int_\mathcal{A} \|Q^*(\cdot, a)\|_{L^2(\mathcal{S}, \mu)}^2 \|f\|_{L^2(\mathcal{S}, \mu)}^2 d\nu(a) \quad \because \text{Cauchy-Schwarz} \\
   &\leq V_{\max}^2 \mu(\mathcal{S})\nu(\mathcal{A})\|f\|_{L^2(\mathcal{S}, \mu)}^2. \qquad \because \|Q^*(\cdot, a)\|_{L^2(\mathcal{S})}^2 \leq V_{\max}^2 \mu(\mathcal{S})
   \end{aligned}$$

   Next, we show that $K : L^2(\mathcal{S}, \mu) \to L^2(\mathcal{A}, \nu)$ is indeed a compact operator. It suffices to show that for any bounded sequence $(f_n)_{n \geq 1}$ in $L^2(\mathcal{S}, \mu)$, the sequence $(Kf_n)_{n \geq 1}$ contains a convergent subsequence. For this, we use (generalized) Arzelà-Ascoli theorem, which states that if $(Kf_n)_{n \geq 1}$ is uniformly bounded and uniformly equicontinuous, then it contains a convergent

subsequence. To that end, first note that $\|Kf_n\| \leq \|K\|\|f_n\|$ and therefore, if $\|f_n\| \leq B$ for all $n \geq 1$, then $\|Kf_n\| \leq \|K\|B$ for all $n \geq 1$. That is, the sequence $(Kf_n)_{n \geq 1}$ is uniformly bounded. Next, we can also verify that $(Kf_n)_{n \geq 1}$ is equicontinuous because for all $n \geq 1$,

$$
\begin{aligned}
\left| Kf_n(a_1) - Kf_n(a_2) \right| &\leq \left| \int_{\mathcal{S}} \{Q^*(s, a_1) - Q^*(s, a_2)\} f_n(s) d\mu(s) \right| \\
&\leq \|Q^*(s, a_1) - Q^*(s, a_2)\|_{L^2(\mathcal{S})} \|f_n\|_{L^2(\mathcal{S}, \mu)} \\
&\leq L\mu(\mathcal{S})^{\frac{1}{2}} d_{\mathcal{A}}(a_1, a_2) \|f_n\|_{L^2(\mathcal{S}, \mu)} \\
&\leq BL\mu(\mathcal{S})^{\frac{1}{2}} d_{\mathcal{A}}(a_1, a_2).
\end{aligned}
$$

In the second to last inequality, we used the fact that $Q^*$ is $L$-Lipschitz to show

$$
\begin{aligned}
\|Q^*(s, a_1) - Q^*(s, a_2)\|_{L^2(\mathcal{S}, \mu)} \|f_n\|_{L^2(\mathcal{S}, \mu)} &= \left( \int_{\mathcal{S}} \left( Q^*(s, a_1) - Q^*(s, a_2) \right)^2 d\mu(S) \right)^{\frac{1}{2}} \\
&\leq \left( \int_{\mathcal{S}} L^2 d_{\mathcal{A}}(a_1, a_2)^2 d\mu(S) \right)^{\frac{1}{2}} \\
&= L\mu(\mathcal{S})^{\frac{1}{2}} d_{\mathcal{A}}(a_1, a_2).
\end{aligned}
$$

2. Spectral decomposition of $K$.

- First of all, we show that there exist orthonormal bases $\{f_i \in L^2(\mathcal{S}, \mu) : i \in \mathbb{N}\}$, $\{g_i \in L^2(\mathcal{A}, \nu) : i \in \mathbb{N}\}$ and singular values $\{\sigma_i \geq 0 : i \in \mathbb{N}\}$ such that

$$
K = \sum_{i=1}^{\infty} \sigma_i f_i^* \otimes g_i. \tag{9}
$$

To see this, we consider the adjoint operator of $K$, namely, $K^* : L^2(\mathcal{A}, \nu) \to L^2(\mathcal{S}, \mu)$. Since $K : L^2(\mathcal{S}, \mu) \to L^2(\mathcal{A}, \nu)$ is compact, $K^*$ is also compact. Note that $K^*K$ is compact and self-adjoint. By the spectral theorem for compact self-adjoint operators, there exist $\{\tau_i \in \mathbb{R} : i \in \mathbb{N}\}$ and an orthonormal basis $\{f_i \in L^2(\mathcal{S}, \mu) : i \in \mathbb{N}\}$ such that $K^*Kf_i = \tau_i f_i$ for all $i \in \mathbb{N}$. We can observe that $\tau_i \geq 0$ for all $i$ because $\tau_i = \tau_i \langle f_i, f_i \rangle = \langle K^*Kf_i, f_i \rangle = \|Kf_i\|_{L^2(\mathcal{S}, \mu)}^2 \geq 0$. We let $I := \{i \in \mathbb{N} : \tau_i > 0\}$.
Next, we observe that $\ker(K^*K) = \ker(K)$. Showing $\ker(K^*K) \supseteq \ker(K)$ is trivial. To show the other direction, let's suppose that $f \in \ker(K^*K)$. Then $\|Kf\|_{L^2(\mathcal{A}, \nu)}^2 = \langle Kf, Kf, \rangle = \langle K^*Kf, f \rangle = 0$, which requires $Kf = 0$ and thus $f \in \ker(K)$.
For $i \in I$, we let $g_i = \frac{1}{\sqrt{\tau_i}} Kf_i$. Then $\langle g_i, g_j \rangle = \frac{1}{\sqrt{\tau_i \tau_j}} \langle Kf_i, Kf_j \rangle = \frac{1}{\sqrt{\tau_i \tau_j}} \langle K^*Kf_i, f_j \rangle = \delta_{ij}$, and hence, $\{g_i : i \in I\}$ consists of orthonormal vectors. We can augment $\{g_i : i \in I\}$ by adding appropriate vectors to make $\{g_i : i \in \mathbb{N}\}$ an orthonormal basis of $L^2(\mathcal{A}, \nu)$.
Every vector $\phi \in L^2(\mathcal{S}, \mu)$ can be expanded as $\phi = \sum_{i=1}^{\infty} \langle \phi, f_i \rangle f_i$. Then we see that $K\phi = \sum_{i=1}^{\infty} \langle \phi, f_i \rangle Kf_i = \sum_{i=1}^{\infty} \sqrt{\tau_i} \langle \phi, f_i \rangle g_i$. By letting $\sigma_i = \sqrt{\tau_i}$, we obtain (9).

- In addition, we show that $\sum_{i=1}^{\infty} \sigma_i^2 = \|Q^*\|_{L^2(\mathcal{S} \times \mathcal{A}, \mu \times \nu)}^2 < \infty$. The Hilbert-Schmidt norm of operator $K$ is defined as $\|K\|_{HS} = \text{Tr}(K^*K) = \sum_{i=1}^{\infty} \|Kf_i\|_{L^2(\mathcal{A}, \nu)}^2 < \infty$. Note that $\|K\|_{HS} = \sum_{i=1}^{\infty} \sigma_i^2$.
First, we observe that for each $i \in \mathbb{N}$,

$$
\begin{aligned}
\langle Kf_i, Kf_i \rangle_{L^2(\mathcal{A}, \nu)} &= \int_{\mathcal{A}} \left( \int_{\mathcal{S}} Q^*(s, a) f_i(s) d\mu(s) \right)^2 d\nu(a) \\
&= \int_{\mathcal{A}} \langle Q^*(\cdot, a), f_i \rangle_{L^2(\mathcal{S}, \mu)}^2 d\nu(a).
\end{aligned}
$$

We define a function $G(a) := \langle Q^*(\cdot, a), f_i \rangle_{L^2(\mathcal{S}, \mu)}^2$. Recall that $Q^* \in L^2(\mathcal{S} \times \mathcal{A}, \mu \times \nu)$ and observe that $G$ is a nonnegative measurable function. Then we can use Tonelli's theorem to see

that

$$\text{Tr}(K^*K) = \sum_{i=1}^{\infty} \langle Kf_i, Kf_i \rangle_{L^2(\mathcal{A},\nu)} = \sum_{i=1}^{\infty} \int_{\mathcal{A}} \langle Q^*(\cdot,a), f_i \rangle^2_{L^2(\mathcal{S},\mu)} d\nu(a)$$

$$= \int_{\mathcal{A}} \sum_{i=1}^{\infty} \langle Q^*(\cdot,a), f_i \rangle^2_{L^2(\mathcal{S},\mu)} d\nu(a) \qquad \because \text{ Tonelli's theorem}$$

$$= \int_{\mathcal{A}} \|Q^*(\cdot,a)\|^2_{L^2(\mathcal{S},\mu)} d\nu(a). \qquad \because \text{ the orthonormality of } \{f_i\}$$

We have $\int_{\mathcal{A}} \|Q^*(\cdot,a)\|^2_{L^2(\mathcal{S},\mu)} d\nu(a) = \int_{\mathcal{A}} \left( \int_{\mathcal{S}} |Q^*(s,a)|^2 d\mu(s) \right) d\nu(a) = \|Q^*\|^2_{L^2(\mathcal{S}\times\mathcal{A},\mu\times\nu)}$ by Fubini's theorem and therefore, $\sum_{i=1}^{\infty} \sigma_i^2 = \|Q^*\|^2_{L^2(\mathcal{S}\times\mathcal{A},\mu\times\nu)}$.

3. Spectral decomposition of $Q^*$.

Now we show that $Q^* = \sum_{i=1}^{\infty} \sigma_i f_i \otimes g_i$ for the same singular values $\{\sigma_i \geq 0 : i \in \mathbb{N}\}$ and orthonormal bases $\{f_i \in L^2(\mathcal{S},\mu) : i \in \mathbb{N}\}$, $\{g_i \in L^2(\mathcal{A},\nu) : i \in \mathbb{N}\}$ as in (9).

For that purpose, we assume that

$$Q^* = \sum_{i=1}^{\infty} \sigma_i f_i \otimes g_i + \varepsilon \qquad (10)$$

for some $\varepsilon \in L^2(\mathcal{S} \times \mathcal{A}, \mu \times \nu)$. For all $\phi \in L^2(\mathcal{S},\mu)$ and $\psi \in L^2(\mathcal{A},\nu)$, we have

$$\langle \psi, K\phi \rangle_{L^2(\mathcal{A},\nu)} = \int_{\mathcal{A}} \psi(a) \left( \int_{\mathcal{S}} Q^*(s,a)\phi(s)d\mu(s) \right) d\nu(a)$$

$$= \int_{\mathcal{A}} \psi(a) \left( \int_{\mathcal{S}} \left( \sum_{i=1}^{\infty} \sigma_i f_i(s) g_i(a) + \varepsilon(s,a) \right) \phi(s) d\mu(s) \right) d\nu(a)$$

$$= \int_{\mathcal{A}} \psi(a) \left\langle \sum_{i=1}^{\infty} \sigma_i f_i, \ \phi \right\rangle_{L^2(\mathcal{S},\mu)} g_i(a) d\nu(a)$$

$$\qquad + \int_{\mathcal{A}} \psi(a) \left( \int_{\mathcal{S}} \varepsilon(s,a)\phi(s)d\mu(s) \right) d\nu(a).$$

When $\phi = f_i$ and $\psi = g_j$, we have $\langle g_j, Kf_i \rangle_{L^2(\mathcal{A},\nu)} = \sigma_i \delta_{ij}$. By Fubini's theorem,

$$\sigma_i \delta_{ij} = \sigma_i \langle g_j, g_i \rangle + \int_{\mathcal{S}\times\mathcal{A}} \varepsilon(s,a) f_i(s) g_j(a) d(\mu \times \nu)(s \times a) \qquad (11)$$

$$= \sigma_i \delta_{ij} + \langle \varepsilon, f_i \otimes g_j \rangle_{L^2(\mathcal{S}\times\mathcal{A},\mu\times\nu)}. \qquad (12)$$

In order to satisfy (11), we must have $\langle \varepsilon, f_i \otimes g_j \rangle_{L^2(\mathcal{S}\times\mathcal{A},\mu\times\nu)} = 0$ for all $(i,j) \in \mathbb{N}^2$.

It is known that $L^2(\mathcal{S} \times \mathcal{A}, \mu \times \nu)$ is isomorphic to $L^2(\mathcal{S},\mu) \otimes L^2(\mathcal{A},\nu)$ and $\{f_i \otimes g_j : (i,j) \in \mathbb{N}^2\}$ constitutes an orthonormal basis of $L^2(\mathcal{S},\mu) \otimes L^2(\mathcal{A},\nu)$. Therefore, $\varepsilon = 0$ and $Q^* = \sum_{i=1}^{\infty} \sigma_i f_i \otimes g_i$.

4. Best rank-$r$ approximation of $Q^*$.

Without loss of generality, we may assume $\sigma_1 \geq \sigma_2 \geq \cdots \geq 0$, i.e., the singular values are sorted in descending order. For any finite $r \in \mathbb{N}$, let $Q^*_r = \sum_{i=1}^{r} \sigma_i f_i \otimes g_i$.

Then,

$$\|Q^* - Q^*_r\|^2_{L^2(\mathcal{S}\times\mathcal{A},\mu\times\nu)} = \left\| \sum_{i=r+1}^{\infty} \sigma_i f_i \otimes g_i \right\|^2_{L^2(\mathcal{S}\times\mathcal{A},\mu\times\nu)}$$

$$= \sum_{i,j=r+1}^{\infty} \sigma_i \sigma_j \langle f_i \otimes g_i, f_j \otimes g_j \rangle_{L^2(\mathcal{S}\times\mathcal{A},\mu\times\nu)}$$

$$= \sum_{i=r+1}^{\infty} \sigma_i^2$$

where we have used the orthonormality of $\{f_i\}$ and $\{g_i\}$.

We conclude the proof with two final remarks:

- Among all rank-$r$ functions of the form $\sum_{i=1}^{r} \lambda_i \phi_i \otimes \psi_i$ for some $\phi_i \in L^2(\mathcal{S}, \mu)$, $\psi_i \in L^2(\mathcal{A}, \nu)$, $Q_r^*$ is the "best" rank-$r$ approximation of $Q^*$ in the $L^2(\mathcal{S} \times \mathcal{A}, \mu \times \nu)$ sense.
- Since $\sum_{i=1}^{\infty} \sigma_i^2 < \infty$, for any $\delta > 0$, there exists $r = r(\delta)$ so that $\sum_{i=r+1}^{\infty} \sigma_i^2 < \delta$. That is, we can approximate $Q_r^*$ arbitrarily well with a sufficiently large, yet still finite, rank $r$.

This completes the proof of Theorem 7. $\qquad\square$

## C   Proof of Theorem 2

### C.1   Helper Lemma: Error Bound for Lookahead Subroutine

This section is devoted to the proof of Theorem 2. To this end, we first need to understand the error guarantees for the lookahead (exploration) subroutine based on the current oracle $V^{(t-1)}$, cf. Eq. (3) and Line 8 of Algorithm 1. This is summarized in the following lemma.

**Lemma 8.** *Suppose that we have access to a value oracle $V : \mathcal{S} \to \mathbb{R}$ such that*

$$\sup_{s \in \mathcal{S}} \big|V(s) - V^*(s)\big| \leq B.$$

*Given $(s, a) \in \mathcal{S} \times \mathcal{A}$, let $s_1', \ldots, s_N'$ be the next states of $(s, a)$ independently drawn from the generative model and let $\hat{Q}(s, a) = R(s, a) + \gamma \cdot \frac{1}{N} \sum_{i=1}^{N} V(s_i')$. Then for any $\delta > 0$,*

$$|\hat{Q}(s, a) - Q^*(s, a)| \leq \gamma \left( B + \sqrt{\frac{2V_{\max}^2}{N} \log\left(\frac{2}{\delta}\right)} \right)$$

*with probability at least $1 - \delta$.*

*Proof.* Note that $Q^*(s, a) = R(s, a) + \gamma \mathbb{E}_{s' \sim P_{s,a}}[V^*(s')]$ by definition of $Q^*$ and $V^*$ (cf. Bellman equation). It follows that

$$
\begin{aligned}
|\hat{Q}(s, a) - Q^*(s, a)| &= \gamma \left| \frac{1}{N} \sum_{i=1}^{N} V(s_i') - \mathbb{E}_{s' \sim P_{s,a}}[V^*(s')] \right| \\
&\leq \gamma \left| \frac{1}{N} \sum_{i=1}^{N} V(s_i') - \frac{1}{N} \sum_{i=1}^{N} V^*(s_i') \right| + \gamma \left| \frac{1}{N} \sum_{i=1}^{N} V^*(s_i') - \mathbb{E}_{s' \sim P_{sa}}[V^*(s')] \right| \\
&= \frac{\gamma}{N} \sum_{i=1}^{N} \big|V(s_i') - V^*(s_i')\big| + \gamma \left| \frac{1}{N} \sum_{i=1}^{N} V^*(s_i') - \mathbb{E}_{s' \sim P_{sa}}[V^*(s')] \right|. \quad (13)
\end{aligned}
$$

By assumption, the first term in Eq. (13) is bounded by $\gamma B$. Meanwhile, since $|V^*(s')| \leq V_{\max}$, we can apply Hoeffding's inequality to control the second term. Specifically, for any $t > 0$,

$$\Pr\left( \frac{1}{N} \sum_{i=1}^{N} V^*(s_i') - \mathbb{E}_{s' \sim P_{sa}}[V^*(s')] > t \right) \leq \exp\left( -\frac{Nt^2}{2V_{\max}^2} \right).$$

Solving $\delta = 2\exp\left( -\frac{Nt^2}{2V_{\max}^2} \right)$ for $t$ yields $t = \sqrt{\frac{2V_{\max}^2}{N} \log\left(\frac{2}{\delta}\right)}$ and this completes the proof. $\quad\square$

### C.2   Proof of Theorem 2

*Proof of Theorem 2.* We prove the first statement by mathematical induction. For $t = 0$, $Q^{(0)}(s, a) \equiv 0$ and thus $|Q^{(0)}(s, a) - Q^*(s, a)| \leq V_{\max}$ for all $(s, a)$. Next, we want to show that for $t = 1, \ldots, T$,

$$\sup_{(s,a) \in \mathcal{S} \times \mathcal{A}} \big|Q^{(t)}(s, a) - Q^*(s, a)\big| \leq \rho \sup_{(s,a) \in \mathcal{S} \times \mathcal{A}} \big|Q^{(t-1)}(s, a) - Q^*(s, a)\big|. \quad (14)$$

Fix $t$ and suppose that $\sup_{(s,a)\in\mathcal{S}\times\mathcal{A}} \left|Q^{(t-1)}(s,a) - Q^*(s,a)\right| \leq B^{(t-1)}$. Note that this implies $\sup_{s\in\mathcal{S}} \left|V^{(t-1)}(s) - V^*(s)\right| \leq B^{(t-1)}$ because $Q^{(t-1)}, Q^*$ are continuous and $\mathcal{A}$ is compact [2]. To prove the inequality in Eq. (14), we backtrack the updating steps in Algorithm 1.

For each $s \in \mathcal{S}$ and $a \in \mathcal{A}$, let $\hat{s}^{(t)} \in \arg\min_{s'\in\mathcal{S}^{(t)}} \|s'-s\|_2$ and $\hat{a}^{(t)} \in \arg\min_{a'\in\mathcal{A}^{(t)}} \|a'-a\|_2$. Since $\mathcal{S}^{(t)}$ is a $\beta^{(t)}$-net of $\mathcal{S}$, $\|\hat{s}^{(t)} - s\| \leq \beta^{(t)}$. Likewise, $\|\hat{a}^{(t)} - a\| \leq \beta^{(t)}$. As $Q^{(t)}(s,a) = \bar{Q}^{(t)}(\hat{s}^{(t)}, \hat{a}^{(t)})$ and $Q^*$ is $L$-Lipschitz,

$$
\begin{aligned}
\left|Q^{(t)}(s,a) - Q^*(s,a)\right| &= \left|\bar{Q}^{(t)}(\hat{s}^{(t)}, \hat{a}^{(t)}) - Q^*(s,a)\right| \\
&= \left|\bar{Q}^{(t)}(\hat{s}^{(t)}, \hat{a}^{(t)}) - Q^*(\hat{s}^{(t)}, \hat{a}^{(t)})\right| + \left|Q^*(\hat{s}^{(t)}, \hat{a}^{(t)}) - Q^*(s,a)\right| \\
&\leq \left|\bar{Q}^{(t)}(\hat{s}^{(t)}, \hat{a}^{(t)}) - Q^*(\hat{s}^{(t)}, \hat{a}^{(t)})\right| + 2L\beta^{(t)}.
\end{aligned}
$$

Therefore, we obtain the following upper bound for Step 4 (interpolation):

$$
\sup_{(s,a)\in\mathcal{S}\times\mathcal{A}} \left|Q^{(t)}(s,a) - Q^*(s,a)\right| \leq \max_{(s,a)\in\mathcal{S}^{(t)}\times\mathcal{A}^{(t)}} \left|\bar{Q}^{(t)}(s,a) - Q^*(s,a)\right| + 2L\beta^{(t)}. \tag{15}
$$

By Assumption 1, we have the following upper bound for Step 3 (matrix estimation):

$$
\max_{(s,a)\in\mathcal{S}^{(t)}\times\mathcal{A}^{(t)}} \left|\bar{Q}^{(t)}(s,a) - Q^*(s,a)\right| \leq \mathsf{c}_{\mathsf{me}} \max_{(s,a)\in\Omega^{(t)}} \left|\hat{Q}^{(t)}(s,a) - Q^*(s,a)\right|. \tag{16}
$$

Lastly, applying Lemma 8 and taking union bound over $(s,a) \in \Omega^{(t)}$, we can show that

$$
\max_{(s,a)\in\Omega^{(t)}} \left|\hat{Q}^{(t)}(s,a) - Q^*(s,a)\right| \leq \gamma \left( B^{(t-1)} + \sqrt{\frac{2V_{\max}^2}{N^{(t)}} \log\left(\frac{2|\Omega^{(t)}|T}{\delta}\right)} \right) \tag{17}
$$

with probability at least $1 - \frac{\delta}{T}$.

Combining Eqs. (15), (16), (17) yields

$$
\sup_{(s,a)\in\mathcal{S}\times\mathcal{A}} \left|Q^{(t)}(s,a) - Q^*(s,a)\right| \leq B^{(t)}
$$

with probability at least $1 - \frac{\delta}{T}$ where

$$
B^{(t)} = \gamma\mathsf{c}_{\mathsf{me}}\left( B^{(t-1)} + \sqrt{\frac{2V_{\max}^2}{N^{(t)}} \log\left(\frac{2|\Omega^{(t)}|T}{\delta}\right)} \right) + 2L\beta^{(t)}.
$$

By Assumption 1, this requires at most $|\Omega^{(t)}| = \mathsf{C}_{\mathsf{me}}\left(|\mathcal{S}^{(t)}| + |\mathcal{A}^{(t)}|\right)$. Moreover, for each $1 \leq t \leq T$, if we choose $\beta^{(t)} = \frac{V_{\max}}{8L}(2\gamma\mathsf{c}_{\mathsf{me}})^t$ and

$$
N^{(t)} = \frac{8}{(2\gamma\mathsf{c}_{\mathsf{me}})^{2(t-1)}} \log\left(\frac{2|\Omega^{(t)}|T}{\delta}\right), \tag{18}
$$

then $B^{(t-1)} \leq (2\gamma\mathsf{c}_{\mathsf{me}})^{t-1}V_{\max}$ implies that $B^{(t)} \leq (2\gamma\mathsf{c}_{\mathsf{me}})^t V_{\max}$ with probability at least $1 - \frac{\delta}{T}$.

At the beginning, we observed $|Q^{(0)}(s,a) - Q^*(s,a)| \leq V_{\max}$ for all $(s,a)$, i.e., $B^{(0)} \leq V_{\max}$. By taking the union bound over $t = 1, \ldots, T$,

$$
\sup_{(s,a)\in\mathcal{S}\times\mathcal{A}} \left|Q^{(t)}(s,a) - Q^*(s,a)\right| \leq (2\gamma\mathsf{c}_{\mathsf{me}})^t V_{\max}, \quad \forall t = 1, \ldots, T
$$

with probability at least $1 - \delta$.

**Sample complexity.** If $\gamma < \frac{1}{2c_{\mathsf{me}}}$, then $2\gamma c_{\mathsf{me}} < 1$. Let $T_\epsilon = \left\lceil \frac{\log\left(\frac{V_{\max}}{\epsilon}\right)}{\log\left(\frac{1}{2\gamma c_{\mathsf{me}}}\right)} \right\rceil$ and observe that $(2\gamma c_{\mathsf{me}})\epsilon \leq (2\gamma c_{\mathsf{me}})^{T_\epsilon} V_{\max} \leq \epsilon$. For each $t, 1 \leq t \leq T$, we query $\hat{Q}^{(t)}(s,a)$ for $(s,a) \in \Omega^{(t)}$, each of which requires exploring $N^{(t)}$ samples. Therefore, the total sample complexity of Algorithm 1 with $T = T_\epsilon$ is $\sum_{t=1}^{T_\epsilon} |\Omega^{(t)}| N^{(t)}$.

By standard argument on covering number, we can see that $|\mathcal{S}^{(t)}|, |\mathcal{A}^{(t)}| \leq C'\left(\frac{1}{\beta^{(t)}}\right)^d = C'\left(\frac{8L}{V_{\max}}\right)^d (2\gamma c_{\mathsf{me}})^{-dt}$ for some absolute constant $C' > 0$. This is an increasing function of $t$ and hence, $|\Omega^{(t)}| = C_{\mathsf{me}}\left(|\mathcal{S}^{(t)}| + |\mathcal{A}^{(t)}|\right)$ and $N^{(t)}$ as described in Eq. (18) are also increasing with respect to $t$.

Observe that $\beta^{(T_\epsilon)} = \frac{V_{\max}}{8L}(2\gamma c_{\mathsf{me}})^{T_\epsilon} \geq \frac{2\gamma c_{\mathsf{me}}}{8L}\epsilon$. Hence, $|\mathcal{S}^{(T_\epsilon)}|, |\mathcal{A}^{(T_\epsilon)}| \leq C'\left(\frac{8L}{2\gamma c_{\mathsf{me}}}\right)^d \frac{1}{\epsilon^d}$. Therefore, the overall number of samples utilized by the algorithm are

$$\sum_{t=1}^{T_\epsilon} |\Omega^{(t)}| N^{(t)} \leq T_\epsilon |\Omega^{(T_\epsilon)}| N^{(T_\epsilon)}$$

$$\leq T_\epsilon \cdot C_{\mathsf{me}}\left(|\mathcal{S}^{(T_\epsilon)}| + |\mathcal{A}^{(T_\epsilon)}|\right) \cdot \frac{8}{(2\gamma c_{\mathsf{me}})^{2(T_\epsilon - 1)}} \log\left(\frac{2C_{\mathsf{me}}\left(|\mathcal{S}^{(T_\epsilon)}| + |\mathcal{A}^{(T_\epsilon)}|\right) T_\epsilon}{\delta}\right)$$

$$\leq T_\epsilon \cdot 2C_{\mathsf{me}} C'\left(\frac{8L}{2\gamma c_{\mathsf{me}}}\right)^d \frac{1}{\epsilon^d} \cdot 8\left(\frac{V_{\max}}{\epsilon}\right)^2 \log\left(\frac{4C_{\mathsf{me}} C' T_\epsilon}{\delta}\left(\frac{8L}{2\gamma c_{\mathsf{me}}}\right)^d \frac{1}{\epsilon^d}\right)$$

$$= 16 C_{\mathsf{me}} C' V_{\max}^2 \left(\frac{8L}{2\gamma c_{\mathsf{me}}}\right)^d \cdot \frac{T_\epsilon}{\epsilon^{d+2}} \cdot \log\left(4C_{\mathsf{me}} C'\left(\frac{8L}{2\gamma c_{\mathsf{me}}}\right)^d \cdot \frac{T_\epsilon}{\epsilon^d} \cdot \frac{1}{\delta}\right). \quad (19)$$

Since $T_\epsilon = \left\lceil \frac{\log\left(\frac{V_{\max}}{\epsilon}\right)}{\log\left(\frac{1}{2\gamma c_{\mathsf{me}}}\right)} \right\rceil = O\left(\log \frac{1}{\epsilon}\right)$, it follows from (19) that the overall sample complexity scales as $O\left(\frac{1}{\epsilon^{d+2}} \log \frac{1}{\epsilon} \cdot \left(\log \frac{1}{\epsilon} + \log \frac{1}{\delta}\right)\right)$. This completes the proof of Theorem 2. $\qquad\square$

## D Rank($Q^*$) = 1

We state Theorem 9 which incorporates implications of Proposition 3 on Theorem 2. The proof of Proposition 3 and Theorem 9 can be found in our full technical report [34].

### D.1 Theorem 9 = Proposition 3 + Theorem 2

**Theorem 9.** *Let $Q^*$ be rank 1. Consider the RL algorithm (cf. Section 3) with the Matrix Estimation method as described in Section 5.1. If $\gamma < \frac{R_{\min}}{14 R_{\max}}$, then the following two statements are true.*

*1. For any $\delta > 0$, we have*

$$\sup_{(s,a)\in\mathcal{S}\times\mathcal{A}} \left|Q^{(t)}(s,a) - Q^*(s,a)\right| \leq \left(\frac{14 R_{\max}}{R_{\min}}\gamma\right)^t V_{\max}, \quad \forall\, 1 \leq t \leq T,$$

*with probability at least $1 - \delta$ by choosing algorithmic parameters $\beta^{(t)}, N^{(t)}$ appropriately.*

*2. Further, given $\epsilon > 0$, it suffices to set $T = \Theta(\log \frac{1}{\epsilon})$ and use $\tilde{O}(\frac{1}{\epsilon^{d+2}} \cdot \log \frac{1}{\delta})$ number of samples to achieve*

$$\mathbb{P}\left(\sup_{(s,a)\in\mathcal{S}\times\mathcal{A}} \left|Q^{(T)}(s,a) - Q^*(s,a)\right| \leq \epsilon\right) \geq 1 - \delta.$$

## E Rank($Q^*$) = $r$

In this section, we state a general version of Proposition 5. Once we have the general version Proposition 10, we state Theorem 11 which incorporates implications of Proposition 10 on Theorem 2. The proof of Proposition 10 and Theorem 11 can be found in our full technical report [34]. Lastly, we discuss corollaries for finite space in Section E.3.

## E.1 A General Version of Proposition 5

**Proposition 10.** *Let $\Omega^{(t)}$ and $\bar{Q}^{(t)}$ as described above. For any $\epsilon \leq \frac{1}{2\sqrt{|\mathcal{S}^\sharp||\mathcal{A}^\sharp|}} \sigma_r\big(Q^*(\mathcal{S}^\sharp, \mathcal{A}^\sharp)\big)$, if $\max_{(s,a) \in \Omega^{(t)}} \big|\hat{Q}^{(t)}(s,a) - Q^*(s,a)\big| \leq \epsilon$, then*

$$\max_{(s,a) \in \mathcal{S}^{(t)} \times \mathcal{A}^{(t)}} \big|\bar{Q}^{(t)}(s,a) - Q^*(s,a)\big|$$

$$\leq \left(6\sqrt{2}\bigg(\frac{\sqrt{|\mathcal{S}^\sharp||\mathcal{A}^\sharp|}}{\sigma_r\big(Q^*(\mathcal{S}^\sharp, \mathcal{A}^\sharp)\big)}\bigg) + 2(1+\sqrt{5})\bigg(\frac{\sqrt{|\mathcal{S}^\sharp||\mathcal{A}^\sharp|}}{\sigma_r\big(Q^*(\mathcal{S}^\sharp, \mathcal{A}^\sharp)\big)}\bigg)^2\right) V_{\max}\epsilon. \quad (20)$$

## E.2 Theorem 11 = Proposition 10 + Theorem 2

We state Theorem 11 that follows as Corollary of Theorem 2 using Proposition 5 (or Proposition 10). Recall that assuming $|\mathcal{S}^\sharp| = |\mathcal{A}^\sharp| = r$, we defined the following quantity

$$c(r; \mathcal{S}^\sharp, \mathcal{A}^\sharp) = \left(6\sqrt{2}\bigg(\frac{r}{\sigma_r(Q^*(\mathcal{S}^\sharp, \mathcal{A}^\sharp))}\bigg) + 2(1+\sqrt{5})\bigg(\frac{r}{\sigma_r(Q^*(\mathcal{S}^\sharp, \mathcal{A}^\sharp))}\bigg)^2\right) V_{\max}$$

in Proposition 5. This is a special case of $c_{\mathsf{me}}$ for $|\mathcal{S}^\sharp| = |\mathcal{A}^\sharp| = r$ that appears in Proposition 10 as the multiplier on the right-hand side of (20). This quantity appears in the following theorem statement to determine the range of $\gamma$ and the convergence rate.

As a matter of fact, our algorithm does not require $|\mathcal{S}^\sharp| = |\mathcal{A}^\sharp| = r$. We present a general theorem for approximate rank-$r$ setup (Theorem 16) in Appendix F in full generality without assuming $|\mathcal{S}^\sharp| = |\mathcal{A}^\sharp| = r$. One can derive a general version of Theorem 11 for $\mathcal{S}^\sharp, \mathcal{A}^\sharp$ beyond $|\mathcal{S}^\sharp| = |\mathcal{A}^\sharp| = r$ from Theorem 16 by letting $\zeta_r = 0$, where $\zeta_r$ is the approximation error between the rank-$r$ approximation of $Q^*$ and the actual $Q^*$. That is, if $Q^*$ is of rank $r$, $\zeta_r = 0$. Parsing our general results briefly, we remark that as long as $|\mathcal{S}^\sharp| = |\mathcal{A}^\sharp| = O(r)$, we achieve the same scaling of sample complexity in terms of the problem dimensions.

**Theorem 11.** *Let $Q^*$ have rank $r$. Consider the RL algorithm (cf. Section 3) with the Matrix Estimation method as described in Section 5.2. If $\gamma \leq \frac{1}{2c(r;\mathcal{S}^\sharp, \mathcal{A}^\sharp)}$, then the following statements hold.*

1. *For any $\delta > 0$, we have*

$$\sup_{(s,a) \in \mathcal{S} \times \mathcal{A}} \big|Q^{(t)}(s,a) - Q^*(s,a)\big| \leq \big(2c(r; \mathcal{S}^\sharp, \mathcal{A}^\sharp)\gamma\big)^t V_{\max}, \quad \text{for all } t = 1, \ldots, T$$

   *with probability at least $1 - \delta$ by choosing algorithmic parameters $\beta^{(t)}, N^{(t)}$ appropriately.*

2. *Further, given $\epsilon > 0$, it suffices to set $T = \Theta(\log \frac{1}{\epsilon})$ and use $\tilde{O}(\frac{1}{\epsilon^{d+2}} \cdot \log \frac{1}{\delta})$ number of samples to achieve*

$$\mathbb{P}\left(\sup_{(s,a) \in \mathcal{S} \times \mathcal{A}} \big|Q^{(T)}(s,a) - Q^*(s,a)\big| \leq \epsilon\right) \geq 1 - \delta.$$

## E.3 Corollaries of Theorem 11

Recall that our algorithm do not demand any special properties of $\mathcal{S}, \mathcal{A}$ except the existence of $\beta^{(t)}$-net, which is the case whenever $\mathcal{S}, \mathcal{A}$ are compact. Also, our analysis is general in the sense that it only requires $\mathcal{S}, \mathcal{A}$ to be compact with finite measures, and $Q^*$ to be $L$-Lipschitz. Therefore, it is not hard to see that our algorithm and analysis are applicable to the case where state or action space is finite, or both. We summarize results below as corollaries of Theorem 11 without proofs.

Before presenting the results, we recall the following quantity defined in Proposition 5:

$$c(r; \mathcal{S}^\sharp, \mathcal{A}^\sharp) = \left(6\sqrt{2}\bigg(\frac{r}{\sigma_r(Q^*(\mathcal{S}^\sharp, \mathcal{A}^\sharp))}\bigg) + 2(1+\sqrt{5})\bigg(\frac{r}{\sigma_r(Q^*(\mathcal{S}^\sharp, \mathcal{A}^\sharp))}\bigg)^2\right) V_{\max},$$

which is a special case of $c_{\mathsf{me}}$ for $|\mathcal{S}^\sharp| = |\mathcal{A}^\sharp| = r$ that appears in Proposition 10 as the multiplier on the right-hand side of (20). This quantity determines the range of $\gamma$ and the convergence rate.

**Continuous** $\mathcal{S} \subset \mathbb{R}^d$ **and Finite** $\mathcal{A}$. In this case, the algorithm only needs to discretize the state space at each iteration. In other words, $\mathcal{A}^{(t)} = \mathcal{A}$, for all $t = 1, \ldots, T$ and $\Omega^{(t)} = \{(s, a) \in \bar{\mathcal{S}}^{(t)} \times \mathcal{A} : s \in \mathcal{S}^\sharp$ or $a \in \mathcal{A}^\sharp\}$. Finally, the generalization step only needs to interpolate the state space $\mathcal{S}$. Let $|\mathcal{S}^\sharp| = |\mathcal{A}^\sharp| = r$. Then, we have the following guarantees as an immediate corollary of Theorem 11:

**Corollary 12.** *Consider the rank-$r$ setting with continuous $\mathcal{S}$ and finite $\mathcal{A}$. Suppose that we run the RL algorithm (cf. Section 3) with the Matrix Estimation method described in Section 5.2. If $\gamma \leq \frac{1}{2c(r;\mathcal{S}^\sharp,\mathcal{A}^\sharp)}$, then the following holds.*

1. *For any $\delta > 0$, we have*

$$\sup_{(s,a)\in\mathcal{S}\times\mathcal{A}} \left|Q^{(t)}(s,a) - Q^*(s,a)\right| \leq \left(2c(r;\mathcal{S}^\sharp,\mathcal{A}^\sharp)\gamma\right)^t V_{\max}, \quad \text{for all } t = 1, \ldots, T$$

   *with probability at least $1 - \delta$ by choosing algorithmic parameters $\beta^{(t)}, N^{(t)}$ appropriately.*

2. *Further, given $\epsilon > 0$, it suffices to set $T = \Theta(\log\frac{1}{\epsilon})$ and use $\tilde{O}(\frac{1}{\epsilon^{d+2}} \cdot \log\frac{1}{\delta})$ number of samples to achieve*

$$\mathbb{P}\left(\sup_{(s,a)\in\mathcal{S}\times\mathcal{A}} \left|Q^{(T)}(s,a) - Q^*(s,a)\right| \leq \epsilon\right) \geq 1 - \delta.$$

**Finite** $\mathcal{S}$ **and Finite** $\mathcal{A}$. Since the spaces are discrete, we have an optimal $Q^*$ being a $|\mathcal{S}| \times |\mathcal{A}|$ matrix. For this special case, the algorithm simply skips the discretization (i.e., $\beta^{(t)} = 0$) and generalization steps at each iteration. In other words, $\mathcal{S}^{(t)} = \mathcal{S}$ and $\mathcal{A}^{(t)} = \mathcal{A}$, for all $t = 1, \ldots, T$, and $\Omega^{(t)} = \{(s,a) \in \mathcal{S} \times \mathcal{A} : s \in \mathcal{S}^\sharp$ or $a \in \mathcal{A}^\sharp\}$. Suppose that the optimal matrix $Q^*(\mathcal{S}, \mathcal{A})$ is rank-$r$ and let $|\mathcal{S}^\sharp| = |\mathcal{A}^\sharp| = r$. We then have the following guarantees:

**Corollary 13.** *Consider finite $\mathcal{S}$ and finite $\mathcal{A}$ with the optimal matrix $Q^*(\mathcal{S}, \mathcal{A})$ being rank-$r$. Suppose that we run the RL algorithm (cf. Section 3) with the Matrix Estimation method described in Section 5.2. If $\gamma \leq \frac{1}{2c(r;\mathcal{S}^\sharp,\mathcal{A}^\sharp)}$, then the following holds.*

1. *For any $\delta > 0$, we have*

$$\sup_{(s,a)\in\mathcal{S}\times\mathcal{A}} \left|Q^{(t)}(s,a) - Q^*(s,a)\right| \leq \left(2c(r;\mathcal{S}^\sharp,\mathcal{A}^\sharp)\gamma\right)^t V_{\max}, \quad \text{for all } t = 1, \ldots, T$$

   *with probability at least $1 - \delta$ by choosing algorithmic parameters $\beta^{(t)}, N^{(t)}$ appropriately.*

2. *Further, given $\epsilon > 0$, it suffices to set $T = \Theta(\log\frac{1}{\epsilon})$ and use $\tilde{O}(\frac{\max(|\mathcal{S}|,|\mathcal{A}|)}{\epsilon^2} \cdot \log\frac{1}{\delta})$ number of samples to achieve*

$$\mathbb{P}\left(\sup_{(s,a)\in\mathcal{S}\times\mathcal{A}} \left|Q^{(T)}(s,a) - Q^*(s,a)\right| \leq \epsilon\right) \geq 1 - \delta.$$

# F   Approximate Rank-$r$ Reinforcement Learning

In Section 5.2, we considered the setup where the underlying $Q^*$ has rank $r$. In practice, the $Q$ function may not have an exact low-rank structure, but it can be well approximated by the first few spectral components in many cases. In this section, we consider such an approximate rank-$r$ setup.

To be precise, for a positive integer $r$, we let $Q_r^*$ denote the best rank-$r$ approximation of $Q^*$ in the $L^2$-sense (cf. Theorem 1 and its generalized version Theorem 7 in Appendix B). As a reminder, by the spectral theorem of compact operators, we can always write $Q^*(s,a) = \sum_{i=1}^\infty \sigma_i f_i(s) g_i(a)$ with $\sigma_1 \geq \sigma_2 \geq \ldots \geq 0$ and $f_i, g_i$ form orthonormal bases in $L^2(\mathcal{S})$ and $L^2(\mathcal{A})$. Therefore, $Q_r^*(s,a) = \sum_{i=1}^r \sigma_i f_i(s) g_i(a)$.

We begin with introducing the notion of $r$-anchor states/actions that generalizes Definition 4.

**Definition 14.** *($r$-Anchor States and Actions) A set of states $\mathcal{S}^\sharp = \{s_i^\sharp\}_{i=1}^{R_s} \subset \mathcal{S}$ and actions $\mathcal{A}^\sharp = \{a_i^\sharp\}_{i=1}^{R_a} \subset \mathcal{A}$ for some $R_s, R_a$ are called $r$-anchor states and $r$-anchor actions for $Q^*$ if $\text{rank } Q_r^*(\mathcal{S}^\sharp, \mathcal{A}^\sharp) = r$ for a positive integer $r$.*

It is easy to see that if $\mathcal{S}^\sharp$ and $\mathcal{A}^\sharp$ are $r$-anchor states/actions for $Q^*$, then they are $r'$-anchor states/actions for $Q^*$ for all $r' \leq r$.

**Matrix Estimation Algorithm.** The algorithm remains the same as the exact rank-$r$ case, except that we replace anchor states and actions with $r$-anchor states and actions. Precisely, we select $r$-anchor states $\mathcal{S}^\sharp \subset \mathcal{S}$, $r$-anchor actions $\mathcal{A}^\sharp \subset \mathcal{A}$ and fix them throughout all iterations $1 \leq t \leq T$. At each iteration, we choose appropriate $\beta^{(t)}$-nets $\mathcal{S}^{(t)}$ and $\mathcal{A}^{(t)}$ and augment them with the $r$-anchor states and $r$-anchors actions: $\bar{\mathcal{S}}^{(t)} \leftarrow \mathcal{S}^{(t)} \cup \mathcal{S}^\sharp$ and $\bar{\mathcal{A}}^{(t)} \leftarrow \mathcal{A}^{(t)} \cup \mathcal{A}^\sharp$. Then, we set the exploration set as $\Omega^{(t)} = \{(s,a) \in \bar{\mathcal{S}}^{(t)} \times \bar{\mathcal{A}}^{(t)} : s \in \mathcal{S}^\sharp \text{ or } a \in \mathcal{A}^\sharp\}$.

Given estimation $\hat{Q}^{(t)}(s,a)$ for $(s,a) \in \Omega^{(t)}$, the ME method produces estimates for all $(s,a) \in \mathcal{S}^{(t)} \times \mathcal{A}^{(t)}$ as

$$\bar{Q}^{(t)}(s,a) = \hat{Q}^{(t)}(s, \mathcal{A}^\sharp)\big[\hat{Q}^{(t)}(\mathcal{S}^\sharp, \mathcal{A}^\sharp)\big]^\dagger \hat{Q}^{(t)}(\mathcal{S}^\sharp, a) \tag{21}$$

where $\big[\hat{Q}^{(t)}(\mathcal{S}^\sharp, \mathcal{A}^\sharp)\big]^\dagger$ is the Moore-Penrose pseudoinverse of $\hat{Q}^{(t)}(\mathcal{S}^\sharp, \mathcal{A}^\sharp)$. Again, with choice of $R_s = |\mathcal{S}^\sharp| = r$ and $R_a = |\mathcal{A}^\sharp| = r$ (or in general, a constant mulitple of $r$), the size of the exploration set is at most $r\big(|\bar{\mathcal{S}}^{(t)}| + |\bar{\mathcal{A}}^{(t)}| - r\big) \ll |\bar{\mathcal{S}}^{(t)}||\bar{\mathcal{A}}^{(t)}|$.

**Theoretical Guarantee.** Previously, we imposed some regularity assumptions on $Q^*$, but the truncated function $Q^*_r$ is not guaranteed to inherit the regularity properties. Here, we additionally assume that (i) $\|Q^*_r\|_\infty \leq V_{\max}$ and (ii) $Q^*_r$ is $L$-Lipschitz, for the convenience of exposition.

At a high level, our analysis is simple: for a given parameter $r > 0$, we treat $Q^*_r$ as the true function and repeat our analysis for the rank-$r$ setup. Of course, there will be an additional bias, $Q^*_r(s,a) - Q^*(s,a)$, incurred by this substitution which requires careful tracking at each iteration. We formalize this argument in the following proposition and theorem.

**Proposition 15.** *Let $\Omega^{(t)}$ and $\bar{Q}^{(t)}$ as described above. Given a positive integer $r > 0$, let $\mathcal{S}^\sharp$ and $\mathcal{A}^\sharp$ be some $r$-anchor states and actions for $Q^*$.*

*For any $\epsilon \leq \frac{1}{2\sqrt{|\mathcal{S}^\sharp||\mathcal{A}^\sharp|}} \sigma_r\big(Q^*_r(\mathcal{S}^\sharp, \mathcal{A}^\sharp)\big)$, if $\max_{(s,a) \in \Omega^{(t)}} \big|\hat{Q}^{(t)}(s,a) - Q^*_r(s,a)\big| \leq \epsilon$, then*

$$\max_{(s,a) \in \mathcal{S}^{(t)} \times \mathcal{A}^{(t)}} \big|\bar{Q}^{(t)}(s,a) - Q^*_r(s,a)\big| \leq \phi_c(r; \mathcal{S}^\sharp, \mathcal{A}^\sharp)\epsilon,$$

*where*

$$\phi_c(r; \mathcal{S}^\sharp, \mathcal{A}^\sharp) = \left(6\sqrt{2}\left(\frac{\sqrt{|\mathcal{S}^\sharp||\mathcal{A}^\sharp|}}{\sigma_r\big(Q^*_r(\mathcal{S}^\sharp, \mathcal{A}^\sharp)\big)}\right) + 2(1+\sqrt{5})\left(\frac{\sqrt{|\mathcal{S}^\sharp||\mathcal{A}^\sharp|}}{\sigma_r\big(Q^*_r(\mathcal{S}^\sharp, \mathcal{A}^\sharp)\big)}\right)^2\right)V_{\max}. \tag{22}$$

We omit the proof of Proposition 15 because it is exactly the same with the proof of Proposition 10 with minor modifications. With Proposition 15 at hand, we can obtain the following theorem as a Corollary of Theorem 2 for the approximate rank-$r$ setup. The theorem guarantees that when the model bias $\|Q^*_r - Q^*\|_\infty$ is sufficiently small, we obtain convergence and sample complexity results similar to the rank-$r$ setting with an additive error induced by the model bias. Denote by $\zeta_r \triangleq \sup_{(s,a) \in \mathcal{S} \times \mathcal{A}} |Q^*_r(s,a) - Q^*(s,a)|$ the approximation error.

**Theorem 16.** *Consider the approximate rank-$r$ setting in this section. Suppose that we run the RL algorithm (cf. Section 3) with the Matrix Estimation method described above. If $\gamma \leq \frac{1}{2\phi_c(r; \mathcal{S}^\sharp, \mathcal{A}^\sharp)}$ and $\zeta_r \leq \min\left\{\frac{\sigma_r\big(Q^*_r(\mathcal{S}^\sharp, \mathcal{A}^\sharp)\big)}{2\sqrt{|\mathcal{S}^\sharp||\mathcal{A}^\sharp|} + (1 + \frac{1}{V_{\max}})\sigma_r\big(Q^*_r(\mathcal{S}^\sharp, \mathcal{A}^\sharp)\big)}, \frac{3}{2}V_{\max}\right\}$, then the following holds:*

*1. For any $\delta > 0$, with probability at least $1 - \delta$, the following inequality holds for all $t = 1, \ldots, T$*

$$\sup_{(s,a) \in \mathcal{S} \times \mathcal{A}} \big|Q^{(t)}(s,a) - Q^*(s,a)\big| \leq \big(2\phi_c(r; \mathcal{S}^\sharp, \mathcal{A}^\sharp)\gamma\big)^t V_{\max}$$

$$+ (1 + \phi_c(r; \mathcal{S}^\sharp, \mathcal{A}^\sharp)\gamma)\zeta_r \sum_{i=1}^{t} \big(\phi_c(r; \mathcal{S}^\sharp, \mathcal{A}^\sharp)\gamma\big)^{i-1}$$

*by choosing algorithmic parameters $\beta^{(t)}, N^{(t)}$ appropriately.*

2. *Further, given $\epsilon > 0$, it suffices to set $T = \Theta(\log \frac{1}{\epsilon})$ and use $\tilde{O}(\frac{1}{\epsilon^{d+2}} \cdot \log \frac{1}{\delta})$ number of samples to achieve*

$$\mathbb{P}\left( \sup_{(s,a) \in \mathcal{S} \times \mathcal{A}} \left| Q^{(T)}(s,a) - Q^*(s,a) \right| \leq \epsilon + \frac{1 + \gamma \phi_c(r; \mathcal{S}^\sharp, \mathcal{A}^\sharp)}{1 - \gamma \phi_c(r; \mathcal{S}^\sharp, \mathcal{A}^\sharp)} \zeta_r \right) \geq 1 - \delta.$$

The proof of Theorem 16 can be found in the technical report [34]. Theorem 16 establishes the robustness of our method. When the approximation error $\zeta_r$ is not too large, with high probability, we obtain estimate of $Q^*$ that is within $\ell_\infty$ error $\epsilon + \frac{1+\gamma\phi_c(r;\mathcal{S}^\sharp,\mathcal{A}^\sharp)}{1-\gamma\phi_c(r;\mathcal{S}^\sharp,\mathcal{A}^\sharp)}\zeta_r$. Again, the algorithm only efficiently utilizes $\tilde{O}(\frac{1}{\epsilon^{d+2}} \cdot \log \frac{1}{\delta})$ number of samples. Overall, the results on the approximate rank-r setting justifies the soundness of our approach, from both theoretical and practical perspectives.

# G   Additional Discussions on RL and ME

## G.1   Reinforcement Learning

Our work is motivated by the need to improve efficiency of RL algorithms for problems with continuous state and action space, where literature results are scarce. As a byproduct of our analysis, the resulting "low-rank" algorithm can also be reduced to settings where one of the spaces is finite or both. We offer a high-level comparison in Table 1 with a few selected work from literature to help readers see how our approach fares with others from literature. This is by no means a complete illustration, given the vast literature on the finite settings.

We remark that Table 1 is not aimed at a strict comparison on sample complexity since each work focuses on different problem settings. Rather, we intend to convey a rough sense of how our efficient algorithm performs in the setting with finite spaces, and especially what we gain in sample complexity with exploiting low-rank structure. In continuous state and action, our algorithm effectively removes the dependence on the smaller dimension by leveraging the low-rank factorization. The same heuristic in fact carries over to the finite cases, where the dependence on the size of smaller space is "removed," i.e., the sample complexity depends on $|\mathcal{S}|$ instead of $|\mathcal{S}||\mathcal{A}|$, assuming $|\mathcal{S}| \geq |\mathcal{A}|$. That is, exploitation of low-rank structure consistently benefits the sample complexity of our method in the same manner for all three settings.

In Table 1, we include two work per setting selected from literature (except the setting with continuous $\mathcal{S}$ & continuous $\mathcal{A}$ where we were not able to find an appropriate work to compare with). This is because there are extremely various problem settings considered in literature, which involve different technical conditions, partly due to the long history involving finite spaces. For example, between the two work selected for continuous $\mathcal{S}$ and finite $\mathcal{A}$, [35] considers learning the $Q$-function in a single sample path, whereas [51] considers learning the $Q$-function with sparse neural networks when re-sampling i.i.d. transitions is possible. Learning from a single sample path is harder than the other setup, and hence, leads to a sample complexity of $\tilde{O}(\frac{1}{\epsilon^{d+3}})$

For problems with finite $\mathcal{S}$ and finite $\mathcal{A}$, there has been a great effort in learning an $\epsilon$-optimal policy instead of just learning an $\epsilon$-optimal value function. In this context, a line of work [37, 38] attempted to improve the dependence on the term $1/(1-\gamma)$ in sample complexity and recently this question is addressed in [37] by achieving an $\tilde{O}(\frac{|\mathcal{S}||\mathcal{A}|}{(1-\gamma)^3\epsilon^2})$ upper bound that matches the lower bound from [3]. Regardless, traditional results on learning $\epsilon$-optimal policy/value commonly scale as the product $|\mathcal{S}||\mathcal{A}|$. The main message we want to convey with Table 1 in the setting is that the dependence of sample complexity on the size of state/action space can be significantly improved from $|\mathcal{S}||\mathcal{A}|$ to $\max\{|\mathcal{S}|, |\mathcal{A}|\}$ by exploiting the low-rank structure of $Q$-function.

## G.2   Matrix Estimation

We analyze the performance of our proposed algorithm in a decoupled fashion, controlling the worst-case error (for a high-probability event w.r.t. the randomness in sampling step). For the success of our analysis, it is imperative for the matrix estimation subroutine to satisfy Assumption 1 with the two constants $C_{me}, c_{me}$ as small as possible. The assumption ensures that the matrix estimation method in use does not amplify the $\ell_\infty$ error too wildly.

**Why Existing Methods Fail.** Matrix estimation has been a popular topic of active research for the last few decades, which culminated in the low-rank matrix completion via convex relaxation of rank minimization [32, 8, 9]. Also, various algorithms for matrix completion/estimation – including singular value thresholding [22, 10] and nuclear-norm regularization [8, 9, 23] – have been proposed and analyzed with provable guarantees. Despite the huge success in both theory and practice, the available analysis for those existing methods only provides a handle on the error measured in Frobenius norm and a few other limited class of norms (Schatten norms, regularizing norm and its dual, etc.) under certain circumstances (Chapters 9-10, [47]). In particular, there are no satisfactory results so far that provide a control on the $\ell_\infty$ error of matrix estimation, to the best of our knowledge.

Recently, the convergence guarantees for the so-called Burer-Monteiro approach, which takes low-rank factor matrices as decision variables (also commonly referred to as "nonconvex optimization" in literature), has been actively studied in pursuit of developing a computationally more efficient alternative of convex program-based approaches [19, 13]. For example, [13] provides an $\ell_\infty$ guarantee under certain setup. However, they assumed i.i.d. zero-mean noise and requires a proper initialization at the ground truth (for analysis). As a result, we were not able to use existing ME methods and their analysis in this work.

We do not believe this is an algorithmic failure of the ME methods, but it is rather a limitation stemming from the disparity between the traditional analysis in ME and the needs in RL application. For example, considering the error in Frobenius norm is natural in the ME tradition for several reasons, but that analysis is not sufficient for applications where entrywise error is more important. Moreover, it seems manageable, but is not straightforward at once how the mathematical conditions for matrix recovery in ME literature will translate in the context of reinforcement learning. For example, the finite-dimensional incoherence condition between the principal subspaces and the measurement in matrix estimation could translate to a similar infinite-dimensional version of incoherence condition, but some efforts would be needed to reforge existing ME analysis to fit in RL applications seamlessly.

**Why Our ME Method Works.** Instead, we develop an alternative matrix estimation subroutine, which is simple, yet sufficiently powerful for our RL task, thereby enabling us to achieve the ultimate conclusion for the RL problem of interest. The proposed method is amenable to $\ell_\infty$-error analysis facilitated by matrix algebra (see Proposition 10 and its proof). At first glance, our proposal seems extremely simple, and one might doubt its efficacy, worrying about its numerical stability, etc. because it involves the pseudoinverse of a matrix. That concern is partly true, but indeed, there are two key factors that make our method work for the problem of our interest.

First, we assume the existence of "anchor" states and actions, which contain all necessary information for the global recovery of $Q^*$. From a theoretical point of view, this assumption is related to the eigengap condition and the incoherence condition between eigenspace and the sampling operator, which are commonly assumed in existing ME literature. From a practical perspective, this means the existence of faithful representatives that reflect the "diversity" of states and actions, which is the case in many real-world applications.

Second, we are not only passively fed with data, but can actively decide which data to collect. Note that our algorithm requires full measurement for the two cylinders (rectangles when represented as a matrix) corresponding to the anchor states and anchor actions without any missing values in them. This is feasible by adaptive sampling, which is not achievable by random sampling. As a byproduct, active sampling allows us to get rid of the spurious log term that appears in sample complexity of existing ME methods as a result of random sampling.

All in all, our ME method is expected to perform reasonably well in the setup considered in this work. We confirm this is the case with experiments (see Section 6 and Appendix H).

**Open Questions for Future Work.** We have seen that the proposed ME method is successful in the extremely sample deficient setting where $|\Omega^{(t)}| \asymp \max\{|\mathcal{S}^{(t)}|, |\mathcal{A}^{(t)}|\}$. However, it seems other existing ME methods based on convex programs also work similarly well, which cannot be expained with the current analysis.

As a matter of fact, when the computation cost is ignored, convex-relaxation-based approach is widely accepted as the best one in terms of robustness. This is glimpsed by the evolution of $\ell_\infty$ error in our experiments; unlike the fluctuations observed in our method and soft-impute, the error steadily decreases for the nuclear norm minimization. Also, we believe existing ME methods can perform better as $|\Omega^{(t)}|$ becomes larger. We have observed that our simple ME method is most efficient in

the sample-deficient setting where $|\Omega^{(t)}| \asymp \max\{|\mathcal{S}^{(t)}|, |\mathcal{A}^{(t)}|\}$, but we do not know if the same conclusion will still hold as $|\Omega^{(t)}|$ increases.

Therefore, how to harmonize existing ME methods and the low-rank RL task we consider in this paper would be an exciting open question. This question might be tackled either by devising new proof techniques to obtain stronger error guarantees for existing ME methods or by improving our decoupled error analysis for RL iteration developed in this paper. We believe both directions are promising and it would be a valuable contribution to make progress in either direction.

## H    Experimental Setup for Stochastic Control Tasks

In this section, we formalize the detailed settings for several stochastic control tasks we use. Following previous work [43, 50], we briefly introduce the background for each task, and then present the system dynamics as well as our simulation setting. For consistency, we follow the dynamics setup in [43, 50], while adding additionally a noise term $\mathcal{N}$ to one dimension of the state dynamics.

**General Setup.** We first discretize the state space and the the action space into a fine grid and run standard value iteration to obtain a proxy of $Q^*$. Subsequently, when measuring the $\ell_\infty$ error, we take the max (absolute) difference between our estimate $Q^{(t)}$ and the proxy of $Q^*$ over this fine grid. For the mean error, we use the average of the (absolute) difference over this grid. For anchor states and actions, we simply select $r$ states and $r$ actions that are well separated in their respective space. To do so, we divide the space uniformly into $r$ parts and then select a state/action from each part randomly. We use $r = 10$ in all experiments. For the baseline method used to confirm the improvement of sample complexity for our method (i.e., Figure 2), we simply run the same algorithm described in Section 3 but without the ME step. That is, at each iteration, instead of only explore the set $\Omega^{(t)}$, the baseline method explores the entire discretized set $\mathcal{S}^{(t)} \times \mathcal{A}^{(t)}$. The other algorithmic parameters such as $N^{(t)}$, the number of independent transitions sampled, are kept the same. In terms of the comparison with different Matrix Estimation methods, we note that as mentioned, the sampling procedure is different: traditional methods often work by independently sampling each entry with some fixed probability $p$, while our method explores a few entire rows and columns. We hence control all the ME methods to have the same number of observations (i.e., same size of the exploration set $\Omega^{(t)}$ as ours) at each iteration, but switch to independent sampling for the traditional methods.

**Inverted Pendulum.** In this control task, we aim to balance an inverted pendulum on the equilibrium position, i.e., the upright position [43]. The angle and the angular speed tuple, $(\theta, \dot{\theta})$, describes the system dynamics, which is formulated as follows [42]:

$$\theta := \theta + \dot{\theta}\,\tau,$$
$$\dot{\theta} := \dot{\theta} + \left(\sin\theta - \dot{\theta} + u\right)\tau + \mathcal{N}(\mu, \sigma^2),$$

where $\tau$ is the time interval between decisions, $u$ denotes the input torque on the pendulum, and $\mathcal{N}$ refers to the noise term we added with mean $\mu$ and variance $\sigma$. We formulate the reward function to stabilize the pendulum on an upright pendulum:

$$r(\theta, u) = -0.1u^2 + \exp\left(\cos\theta - 1\right).$$

In the simulation, we limit the input torque in $[-1, 1]$ and set $\tau = 0.3$, $\mu = 0$, and $\sigma = 0.1$. We discretize each dimension of the state space into 50 values, and action space into 1000 values, which forms the discretization of the optimal $Q$-value matrix to be of dimension $2500 \times 1000$.

**Mountain Car.** The Mountain Car problem aims to drive an under-powered car up to a hill [42]. We use the position and the velocity of the car, $(x, \dot{x})$, to describe the physical dynamics of the system. Denote $\mathcal{N}$ as the noise term added, $u$ as the acceleration input on the car, we can express the system dynamics as

$$x := x + \dot{x} + \mathcal{N}(\mu, \sigma^2),$$
$$\dot{x} := \dot{x} - 0.0025\cos\left(3x\right) + 0.001u.$$

We define a reward function that encourages the car to drive up to the top of the hill at $x_0 = 0.5$:

$$r(x) = \begin{cases} 10, & x \geq x_0, \\ -1, & \text{else.} \end{cases}$$

We follow standard settings [50] to limit the input $u \in [-1, 1]$. We choose $\mu = 0$ and $\sigma = 1e^{-3}$. Similarly, the whole state space is discretized into 2500 values, and the action space is discretized into 1000 values, which translates to a discretization of $2500 \times 1000$ for the optimal $Q$-value matrix.

**Double Integrator.** We consider the Double Integrator system [33], where a unit mass brick moves along the $x$-axis on a frictionless surface. The brick is controlled with a horizontal force input $u$, which aims to regulate the brick to $\boldsymbol{x} = [0, 0]^T$ [43]. Similarly, we use the position and the velocity $(x, \dot{x})$ of the brick to describe the physical dynamics:

$$x := x + \dot{x}\,\tau + \mathcal{N}(\mu, \sigma^2),$$
$$\dot{x} := \dot{x} + u\,\tau,$$

where $\mathcal{N}$ is the noise term added. Following [43], we define the reward function using the quadratic cost formulation, which regulates the brick to $\boldsymbol{x} = [0, 0]^T$:

$$r(x, \dot{x}) = -\frac{1}{2}\left(x^2 + \dot{x}^2\right).$$

The input torque is limited to be $u \in [-1, 1]$. We again set $\tau = 0.1$, $\mu = 0$, and $\sigma = 0.1$. Similar to the previous tasks, we obtain a discretization of $2500 \times 1000$ for the optimal $Q$-value matrix, with state space discretized into 2500 values and action space discretized into 1000 values.

**Cart-Pole.** Despite simple tasks with smaller state dimensions, we consider the harder Cart-Pole problem with 4-dimensional state space [4]. The problem consists a pole attached to a cart moving on a frictionless track, aiming to stabilize the pole at the upright stable position. The cart is controlled by a limited force that can be applied to both sides of the cart. To describe the physical dynamics of the Cart-Pole system, we use a 4-element tuple $(\theta, \dot{\theta}, x, \dot{x})$, corresponding to the angle and the angular speed of the pole, and the position and the speed of the cart. The dynamics can be expressed as follows:

$$\ddot{\theta} := \frac{g \sin\theta - \frac{u + ml\dot{\theta}^2 \sin\theta}{m_c + m} \cos\theta}{l\left(\frac{4}{3} - \frac{m \cos^2\theta}{m_c + m}\right)},$$

$$\ddot{x} := \frac{u + ml\left(\dot{\theta}^2 \sin\theta - \ddot{\theta}\cos\theta\right)}{m_c + m},$$

$$\theta := \theta + \dot{\theta}\,\tau,$$
$$\dot{\theta} := \dot{\theta} + \ddot{\theta}\,\tau + \mathcal{N}(\mu, \sigma^2),$$
$$x := x + \dot{x}\,\tau,$$
$$\dot{x} := \dot{x} + \ddot{x}\,\tau,$$

where $u \in [-10, 10]$ denotes the input applied to the cart, $\mathcal{N}$ with $\mu = 0$ nad $\sigma = 0.1$ denotes the noise term, $m_c = 1kg$ denotes the mass of the cart, $m = 0.1kg$ denotes the mass of the pole, and $g = 9.8m/s^2$ corresponds to the gravity acceleration.

We define the reward function similar to Inverted Pendulum that tries to stabilize the pole in the upright position:

$$r(\theta) = \cos^4(15\theta).$$

In the simulation, we discretize each dimension of the state space into 10 values, and action space into 1000 values, which forms an optimal $Q$-value function as a matrix of dimension $10000 \times 1000$.

**Acrobot.** Finally, we present the Acrobot swinging up task [43]. The Acrobot is an underactuated two-link robotic arm in the vertical plane (i.e., a two-link pendulum), with only an actuator on the second joint. The goal is to stabilize the Acrobot at the upright position. The equations of motion for the Acrobot can be derived using the method of Lagrange [43]. The physical dynamics of the system is described by the angle and the angular speed of both links, i.e., $(\theta_1, \dot{\theta}_1, \theta_2, \dot{\theta}_2)$. Denote $\tau$ as the time interval, $u$ as the input force on the second joint, $\mathcal{N}$ as the noise term added, the dynamics of

Acrobot can be derived as

$$D_1 := m_1 \left(l_1^2 + l_{c1}^2\right) + m_2 \left(l_1^2 + l_2^2 + l_{c2}^2 + 2l_1 l_{c2} \cos \theta_2\right),$$
$$D_2 := m_2 \left(l_2^2 + l_{c2}^2 + l_1 l_{c2} \cos \theta_2\right),$$
$$\phi_2 := m_2 l_{c2} g \sin\left(\theta_1 + \theta_2\right),$$
$$\phi_1 := -m_2 l_1 l_{c2} \dot{\theta}_2 \left(\dot{\theta}_2 + 2\dot{\theta}_1\right) \sin \theta_2 + \left(m_1 l_{c1} + m_2 l_1\right) g \sin \theta_1 + \phi_2,$$
$$\ddot{\theta}_2 := \frac{u + \frac{D_2}{D_1}\phi_1 - m_2 l_1 l_{c2} \dot{\theta}_1^2 \sin \theta_2 - \phi_2}{m_2(l_2^2 + l_{c2}^2) - \frac{D_2^2}{D_1}},$$
$$\ddot{\theta}_1 := -\frac{D_2 \ddot{\theta}_2 + \phi_1}{D_1},$$
$$\theta_1 := \theta_1 + \dot{\theta}_1 \, \tau,$$
$$\dot{\theta}_1 := \dot{\theta}_1 + \ddot{\theta}_1 \, \tau + \mathcal{N}(\mu, \sigma^2),$$
$$\theta_2 := \theta_2 + \dot{\theta}_2 \, \tau,$$
$$\dot{\theta}_2 := \dot{\theta}_2 + \ddot{\theta}_2 \, \tau,$$

where $l_1 = l_2 = 1m$ are the length of two links, $l_{c1} = l_{c2} = 0.5m$ denote position of the center of mass of both links, $m_1 = m_2 = 1kg$ denote the mass of two links, and $g = 9.8m/s^2$ denotes the gravity acceleration. $u$ corresponds to the input force applied, which is limited by $u \in [-10, 10]$.

Similar to the Inverted Pendulum, we define the reward function that favors the Acrobot to stabilize at the upright unstable fixed point $\boldsymbol{x} = [\pi, 0, 0, 0]^T$:

$$r(\boldsymbol{\theta}, u) = \exp\left(-\cos \theta_1 - 1\right) + \exp\left(-\cos\left(\theta_1 + \theta_2\right) - 1\right).$$

Since the state space of Acrobot is also 4-dimensional, we again discretize each dimension of the state space into 10 values, and action space into 1000 values, which forms discretization of the optimal $Q$-value matrix to be of dimension $10000 \times 1000$.

# I  Additional Results on Stochastic Control Tasks

In this section, we provide additional results on all the 5 tasks. These include plots for sample complexity, error guarantees and visualization of the learned policies.

**Summary of Empirical Results.** We remark that the conclusion remains the same as in the main paper (cf. Section 6). Using our low-rank algorithm with the proposed ME method, the sample complexity is significantly improved as compared with the baselines. For the error guarantees, our ME method is very competitive, both in $\ell_\infty$ and mean error. We again note that our simple method is much more efficient in terms of computational complexity, compared to other ME methods based on optimizations. Finally, the visualization of policies demonstrates that the learned policy, obtained from the output $Q^{(T)}$ is often very close to the policy obtained from $Q^*$, and this leads to the desired behavior in terms of performance metrics, as summarized in Table 3 of the main paper. Overall, these consistent results across various stochastic control tasks confirm the efficacy of our generic low-rank algorithm.

## I.1  Inverted Pendulum

**Sample Complexity and Error Guarantees.** Repeated from the main paper for completeness.

(a) Sample Complexity      (b) Sample Complexity      (c) $\ell_\infty$ Errors      (d) Mean Errors

Figure 3: Empirical results on the Inverted Pendulum control task. In (a) and (b), we show the improved sample complexity for achieving different levels of $\ell_\infty$ error and mean error, respectively. In (c) and (d), we compare the $\ell_\infty$ error and the mean error for various ME methods. Results are averaged across 5 runs for each method.

**Policy Visualization.**

(a) Optimal Policy      (b) Soft-Impute      (c) Nuclear Norm      (d) Ours

Figure 4: Policy visualization of different methods on the Inverted Pendulum control task. The policy is obtained from the output $Q^{(T)}$ by taking $\arg\max_{a \in \mathcal{A}} Q^{(T)}(s, a)$ at each state $s$.

## I.2 Mountain Car

**Sample Complexity and Error Guarantees.**

(a) Sample Complexity      (b) Sample Complexity      (c) $\ell_\infty$ Errors      (d) Mean Errors

Figure 5: Empirical results on the Mountain Car control task. In (a) and (b), we show the improved sample complexity for achieving different levels of $\ell_\infty$ error and mean error, respectively. In (c) and (d), we compare the $\ell_\infty$ error and the mean error for various ME methods. Results are averaged across 5 runs for each method.

**Policy Visualization.**

(a) Optimal Policy      (b) Soft-Impute      (c) Nuclear Norm      (d) Ours

Figure 6: Policy visualization of different methods on the Mountain Car control task. The policy is obtained from the output $Q^{(T)}$ by taking $\arg\max_{a \in \mathcal{A}} Q^{(T)}(s, a)$ at each state $s$.

## I.3 Double Integrator

**Sample Complexity and Error Guarantees.**

(a) Sample Complexity    (b) Sample Complexity    (c) $\ell_\infty$ Errors    (d) Mean Errors

Figure 7: Empirical results on the Double Integrator control task. In (a) and (b), we show the improved sample complexity for achieving different levels of $\ell_\infty$ error and mean error, respectively. In (c) and (d), we compare the $\ell_\infty$ error and the mean error for various ME methods. Results are averaged across 5 runs for each method.

**Policy Visualization.**

(a) Optimal Policy    (b) Soft-Impute    (c) Nuclear Norm    (d) Ours

Figure 8: Policy visualization of different methods on the Double Integrator control task. The policy is obtained from the output $Q^{(T)}$ by taking $\arg\max_{a \in \mathcal{A}} Q^{(T)}(s, a)$ at each state $s$.

## I.4   Cart-Pole

**Sample Complexity and Error Guarantees.**

(a) Sample Complexity    (b) Sample Complexity    (c) $\ell_\infty$ Errors    (d) Mean Errors

Figure 9: Empirical results on the Cart-Pole control task. In (a) and (b), we show the improved sample complexity for achieving different levels of $\ell_\infty$ error and mean error, respectively. In (c) and (d), we compare the $\ell_\infty$ error and the mean error for various ME methods. Results are averaged across 5 runs for each method.

**Policy Visualization.**

(a) Optimal Policy    (b) Soft-Impute    (c) Nuclear Norm    (d) Ours

Figure 10: Policy visualization of different methods on the Cart-Pole control task. The policy is obtained from the output $Q^{(T)}$ by taking $\arg\max_{a \in \mathcal{A}} Q^{(T)}(s, a)$ at each state $s$. Recall that the state space is 4-dimensional. We hence visualize a 2-dimensional slice in the figure.

## I.5   Acrobot

**Sample Complexity and Error Guarantees.**

| (a) Sample Complexity | (b) Sample Complexity | (c) $\ell_\infty$ Errors | (d) Mean Errors |

Figure 11: Empirical results on the Acrobot control task. In (a) and (b), we show the improved sample complexity for achieving different levels of $\ell_\infty$ error and mean error, respectively. In (c) and (d), we compare the $\ell_\infty$ error and the mean error for various ME methods. Results are averaged across 5 runs for each method.

**Policy Visualization.**

| (a) Optimal Policy | (b) Soft-Impute | (c) Nuclear Norm | (d) Ours |

Figure 12: Policy visualization of different methods on the Acrobot control task. The policy is obtained from the output $Q^{(T)}$ by taking $\arg\max_{a \in \mathcal{A}} Q^{(T)}(s, a)$ at each state $s$. Recall that the state space is 4-dimensional. We hence visualize a 2-dimensional slice in the figure.

## I.6 Comparisons on Runtime of Different ME Methods

Nuclear norm minimization is known to be computationally expensive for large matrices. Here, we provide a preliminary result on the runtime of different ME methods in our experiments, demonstrating the computational value of our approach beyond its theoretical guarantees.

Specifically, we calculate the average runtime for one iteration using different ME methods on the Inverted Pendulum task with a $2500 \times 1000$ matrix. We leave other hyper-parameters unchanged, and perform 5 runs for each method. As Table 4 reports, the nuclear norm minimization turns out to be computationally most expensive; in contrast, our method is 40x faster, confirming the computational efficiency of our approach.

Table 4: The runtime comparison of different ME methods for one iteration on the Inverted Pendulum task. Results are averaged across 5 runs for each method.

| ME Method | Soft-Impute | Nuclear norm | **Ours** |
|---|---|---|---|
| Runtime (s) | $41.5 \pm {\scriptstyle 1.7}$ | $76.3 \pm {\scriptstyle 8.2}$ | $\mathbf{1.9} \pm \mathbf{.6}$ |

## I.7 Additional Study on the Discounting Factor $\gamma$

Throughout the empirical study, we follow the literature [43, 50] to use a large discounting factor $\gamma$ (i.e., $0.9$) on several real control tasks. We have demonstrated that the proposed low-rank algorithm can perform well on those settings, confirming the efficacy of our method. Just as a final proof of concept for our theoretical guarantees, we provide in this section an ablation study on the $\ell_\infty$ error with smaller value of $\gamma$. We choose $\gamma = 0.5$ on the Inverted Pendulum control task. note that this affects the reward design and changes the original task. The experiment is only meant to further validate our guarantees.

We show the sample complexity as well as the $\ell_\infty$ errors in Fig. 13. As expected, with a smaller $\gamma$, the convergence is faster. Again, the overall conclusion is consistent with the previous experiments: significant gains on sample complexity are achieved by our efficient algorithm, and the performance of our simple ME method is competitive.

(a) Sample Complexity, $\gamma = 0.5$

(b) $\ell_\infty$ Errors, $\gamma = 0.5$

Figure 13: Empirical results on the Inverted Pendulum control task, with $\gamma = 0.5$. We show the improved sample complexity in (a) and compare the $\ell_\infty$ error for various ME methods in (b).

## Footnotes

[2]For each $s \in \mathcal{S}$, there exist $a^{(t-1)}(s), a^*(s) \in \mathcal{A}$ such that $V^{(t-1)}(s) = Q^{(t-1)}(s, a^{(t-1)}(s))$ and $V^*(s) = Q^*(s, a^*(s))$. If $V^{(t-1)}(s) \geq V^*(s)$, then $V^{(t-1)}(s) - V^*(s) = Q^{(t-1)}(s, a^{(t-1)}(s)) - Q^*(s, a^*(s)) \leq Q^{(t-1)}(s, a^{(t-1)}(s)) - Q^*(s, a^{(t-1)}(s))$. If $V^{(t-1)}(s) < V^*(s)$, then $V^*(s) - V^{(t-1)}(s) = Q^*(s, a^*(s)) - Q^{(t-1)}(s, a^{(t-1)}(s)) \leq Q^*(s, a^*(s)) - Q^{(t-1)}(s, a^*(s))$. Therefore, $|V^{(t-1)}(s) - V^*(s)| \leq \max_{a\in\{a^{(t-1)}(s), a^*(s)\}} \{Q^{(t-1)}(s,a) - Q^*(s,a)\}$.