[Reviews · NeurIPS 2020]

Review 1

Summary and Contributions: This paper studies learning Q-function in reinforcement learning problems with continuous state and action spaces, and where the optimal Q-function is known to be low rank (or approximately so). The main contribution is an algorithm for learning an epsilon-approximate of the Q-function by leveraging low-rank factorization, which enjoys a sample complexity of O(\epsilon^{-2 + max(d_1,d_2)}), where d_1 and d_2 denote the dimension of state and action spaces, respectively. This improves over the lower bound of \Omega(\epsilon^{-d_1-d_2-2}), obtained by looking at the problem through the lens of non-parametric regression. Moreover, when applied to RL problems with finite state and action spaces, the presented algorithm yields a sample complexity scaling with max(S,A), as opposed to lower bound for the unstructured case scaling as SA. The presented algorithm relies on a novel low rank Matrix Estimation method, whose design is necessary to obtain the desired sample complexity.

Strengths: RL with structured state and action spaces is an interesting yet challenging topic. This paper, to the best of my knowledge, happens to be among few works studying how to learn Q-function in continuous MDPs, and where some structure is present. The considered low-rank structure in Q-function makes sense in applications of RL, and is well-motivated. The presented algorithm manages to achieve some gain of exploiting structure in the sample complexity – here being to replace the dimensions of spaces with their maximum. Also, when applied to problems with finite state and action spaces, the algorithm is shown to achieve a clear gain in sample complexity over structure-oblivious counterparts. To perform the low-rank factorization, the algorithm crucially relies on a novel Matrix Estimation method tailored for this problem. Overall the various steps of the presented algorithm (Algorithm 1) intuitively makes sense to me. Moreover, the algorithms seems to be easy and straightforward to implement. Finally, the presented algorithm is examined through extensive numerical experiments, and is shown to be empirically sound.

Weaknesses: Below I list my concerns regarding the setup and reported results. - The generative setup: Perhaps one salient weakness of the paper is the restriction to the generative setup. In the finite case, devising an algorithm for the online setup posed more serious challenges than the generative setup. The restriction of the results to the generative setup hides the price to pay for the need to navigate in the MDP. Could you at least elaborate on explaining the potential difficulties and challenges involved in extending the results to the online case? Could one hope for a similar gain in the sample complexity (over structure-oblivious algorithms)? - Existence of anchor points: The proposed Matrix Estimation method seems to crucially rely on the existence (and availability) of a set of appropriate anchor states and actions. Although it is empirically motivated that it is the case in several RL applications, to me it sounds like a critical assumptions silently made by the authors. To me it is not clear whether the existing assumptions directly imply the existence of such anchor pairs. Please clarify. - In the warm-up example in Sec. 5.1, c_me is proportional to R_max/R_min. There it is assumed that R(s,a)\ge R_min >0, which is a very respective assumption. Even if the result for r=1 would be valid if R_min is redefined to be the minimum non-zero reward, the corresponding c_me could be arbitrarily big, thus resulting in a prohibitive sample complexity. Therefore, unless I am missing something, the constant c_me could be very large in some problems. This then implies both a prohibitive sample complexity (in view of (4)) and a very limited range of \gamma for which the result is valid (in view of \gamma <1/(2c_me) ). Could you please clarify? - In the finite unstructured case, existing results indicate that the sample complexity has to scale as \Omega ((1-\gamma)^{-3}). The paper does not clarify why this is no longer the case in the studied structured case (in both finite and infinite spaces). Could you please explain/motivate.

Correctness: The considered low-rank structure (and its approximate notion) for Q-function sounds technically valid to me. I believe the studied structured RL problem is an interesting and significant problem. The design of the algorithm makes perfect sense to me. I was unable to check all the proofs given the limited review time. The results, however, appear correct to me. I also believe that the empirical evaluation of the presented algorithm is conducted correctly.

Clarity: The paper is overall very well-written and well-organized. In particular, the various steps of the algorithm are described well making the paper easy-to-follow.

Relation to Prior Work: The authors cited most relevant paper (the RL side) that I am aware of. In particular, they clarify the potential similarity between their contributions and existing results. Regarding finite MDPs, I would like to add that [3] presents an algorithm whose sample complexity matches the lower bound. If the space permits, it would be nice and informative to briefly overview structured RL algorithms in the regret minimization setup for when some low-rank structure exists in the MDP.

Reproducibility: Yes

Additional Feedback: The paper is very well-polished. However, I found a few typos reported below: l. 78: space ==> spaces In several places: literature ==> the literature Both \ell_\infty and L^\infty are used to indicate the same notion. For consistency, only one them should be used. ==== AFTER REBUTTAL ==== Thanks for your effort. I have read the other reviews and the authors' response, and I will keep my current score. In agreement with other reviewers, I believe that the paper presents novel algorithmic ideas as well as novel technical tools that could be useful for other RL setups too. It is however a pity that the current analysis excludes the interesting regime of \gamma ~ 1. I would like to ask the authors to clearly highlight this fact in the main text, and provide additional remark explaining the reason behind such a limitation.


Review 2

Summary and Contributions: This work proposes a new, sample efficient RL algorithm to estimate the optimal Q function for continuous state and action spaces. To this end, it introduces a representation of Q* based on a generalized singular value decomposition that decomposes each Lipschitz continuous Q function in an infinite sum of basis functions. Based on this decomposition, the authors propose an algorithm to estimate Q* that proceeds in 4 steps: (i) discretize the S-A space, (ii) build an estimate of Q* at a subset of the points in the discretization, (iii) obtain an accurate estimate of Q* over the whole discretization through matrix estimation, (iv) generalize to the continuous S-A space via interpolation. The authors establish probabilistic convergence results when a matrix estimation algorithm with bounded approximation error in l_infinity norm in presence of observations with arbitrary bounded noise is used. Subsequently, they present such a ME algorithm for Q* with exact or approximate low-rank. Finally, they show their method is effective in continuous control tasks even when using discount factors much higher than those dictated by the theory.

Strengths: - Sound analysis of an important problem in RL: sample efficiency of RL in continuous spaces. - Introduction of a novel ME algorithm that can be applied outside of RL scenarios.

Weaknesses: - The method requires a simulator to be available to collect samples at desired target state-action pairs. While I believe this does not hinder the contribution as the sample complexity result is interesting in its own right, it would be beneficial to mention this earlier in the work (e.g in the introduction) to clarify what are the conditions for this method to be applicable. - This is not really a weakness, rather a clarification. In step 4, you say that you use the nearest neighbor to generalize from the discretization to the continuous space. How does that influence the guarantees for the continuous space given in thm2? Could they be improved by using more sophisticated interpolation? - The experimental section could be a bit clearer. For example, it would be interesting to see some quantitative evaluation supporting your claim that your method is “computationally much more efficient”. Moreover, it would be important to spend a few more words on the baselines. AFTER REBUTTAL After reading the other reviews and the author response, I am convinced the contribution of this paper is relevant. Therefore, I confirm my score.

Correctness: The claims appear to be correct.

Clarity: In general, the paper is well written and easy to follow.

Relation to Prior Work: The relation to prior work is clearly discussed.

Reproducibility: Yes

Additional Feedback:


Review 3

Summary and Contributions: In this paper, the authors focus on reinforcement learning (RL) problems in which the environment is modeled as a Markov decision process (MDP) with continuous state and continuous action spaces. Assuming the availability of a generative model, they develop a novel method to learn the optimal $Q$-function when the function is smooth and has low rank. Under the compact domain, bounded reward, and smoothness assumptions, they first show that, for a given confidence level $\delta$, the optimal $Q$-function can be `approximately' expressed as a finite sum of $r(\delta)$ functions. Using the derived $Q$-function representation, they develop an RL algorithm that utilizes a novel matrix estimation algorithm as a subroutine. Finally, they prove the sample complexity of the developed RL algorithm and demonstrate its performance numerically on various stochastic control problems. The key contributions of the paper are the following. First, the authors provide a spectral representation of the optimal $Q$-function, which allows them to approach the RL problem from a matrix estimation perspective. Second, they develop a novel matrix estimation method, which is used as a subroutine in the proposed RL algorithm, and rigorously analyze its performance under various rank conditions. Third, they analyze the sample complexity of the proposed RL algorithm and show that, when the optimal $Q$-function has low rank, the proposed method introduces an exponential improvement in sample complexity with respect to the existing methods.

Strengths: The paper introduces the idea of using a spectral representation of the optimal $Q$-function to improve the sample complexity of RL algorithms. I believe this new idea is highly promising and might provide inspirations to researchers for developing efficient RL algorithms that perform well on special classes of learning problems. Other strengths of the paper include the clarity of presentation, the rigorousness of the analysis, and the detailed explanations of the weaknesses of the proposed method.

Weaknesses: As the authors mentioned in the paper, the sample complexity results apply only to scenarios in which the discount factor is small. In most practical scenarios, the required condition $\gamma$$<$$1/c_{\text{me}}$ on the discount factor is likely to be violated. Another weakness of the paper is the scenarios considered in the experiments section. For comparison purposes, the authors provide empirical results on stochastic control problems such as inverted pendulum, double-integrator, and cart-pole. However, there are no numerical examples that illustrate the performance of the proposed methods on more realistic and practical planning scenarios.

Correctness: The authors provide proofs for the presented results, but they are highly dense and technical. Even though the techniques used in the proofs make sense, I cannot vouch for their correctness since I only skim through them.

Clarity: The paper is very well-written.

Relation to Prior Work: The authors provide a fairly detailed literature review and emphasize the significance of the presented results by comparing them with the existing work. One of the main contributions of the paper is to exponentially improve the sample complexity bound $\Omega(\frac{1}{\epsilon^{d_1+d_2+2}})$, which, according to the authors' claim, is a classical minimax theory result. Unfortunately, I am not familiar with the above-mentioned complexity bound. The authors provide two references, one of which is a book on nonparametric estimation, for this result. Even though I skimmed through those references, I was not able to find the mentioned complexity bound. Since, the exponential complexity improvement is one of the main contributions of the paper, I suggest the authors to either provide a reference to the theorem that states this result or explicitly show that this bound can be derived from existing results.

Reproducibility: Yes

Additional Feedback: The authors mention that the sample complexity results apply to RL problems in which the discount factor is small. If the authors briefly explain in plain English how the discount factor appears in their proofs and why they need small discount factors to prove their claim, that would be helpful to other researchers to improve the results of this paper. In line 43, $Q$ $\rightarrow$ $Q^{\star}$. Post author response: The response was satisfactory, and my positive evaluation of the paper stands.


Review 4

Summary and Contributions: This paper develops a spectral representation of the Q-function and designs a data-efficient learning algorithm if the Q-function has a low-dimensional structure. A novel component is a new low-rank matrix estimation method, which is shown to achieve theoretical bound in max-norm. Equipped with this matrix estimation, the proposed result is able to achieve improved sample complexity for learning a near optimal Q-function.

Strengths: 1. This paper can be treated as a theoretical justification of the empirical success in Yang et al. (2020) that investigates low-rank Q function with matrix estimation. This result further pushes the boundary in the understanding of low-rank structure of Q-function. Yuzhe Yang, Guo Zhang, Zhi Xu, and Dina Katabi. Harnessing structures for value-based planning and reinforcement learning. In International Conference on Learning Representations (ICLR), 2020. 2. A by-product of the theoretical analysis is a new matrix estimation with max-norm error bound. This is new in the matrix estimation literature when the data are adaptively collected.

Weaknesses: 1. The proposed matrix estimation requires a crucial selection of anchor states and actions. Anchor states and actions contain sufficiently information to recover the low-rank Q function. In the matrix estimation algorithm, the authors assume that these anchor states and anchor actions are given, and in the experiments they pick a few states and actions that are far from each other in their respective metric spaces as the anchor states and actions. It would be more convincing if the authors can rigorously state the anchor selection algorithm and prove this algorithm indeed obtains anchor states and action. Otherwise, the challenge of low-rank matrix estimation just transfers to the finding of anchor states and actions. 2. In the main theory, the discounting factor $\gamma$ needs to be very small. For example, in Theorem 2, $\gamma < 1/(2 c_{me})$ and in Proposition 3 (rank 1 case) $c_{me} = 7 R_{max} / R_{min} > 7$. This implies that $\gamma < 1/14$, which seems to be unreasonably small. I notice that the authors used $\gamma = 0.9$ in all the experiments. More discussions on such inconsistency are needed. 3. As shown in all the five experiments, the nuclear norm matrix estimation works very well in practice. It is comparable or better than the proposed method. Given that nuclear norm is a common routine for low-rank matrix estimation, it would be helpful to provide more justifications on why the proposed new matrix estimation approach is needed. For example, some additional experiments might be added. ######### I have read the rebuttal. My concerns have been addressed.

Correctness: yes

Clarity: yes

Relation to Prior Work: yes

Reproducibility: Yes

Additional Feedback:

[Author Response · NeurIPS 2020]

We thank all the reviewers for their insightful comments and for the acknowledgement of our contributions in this work.
We intend to do our best to incorporate their feedback into our revision. Here we would like to use this opportunity to
clarify some *important aspects* of our work first and respond to the *remaining points* raised by each reviewer.

**1. Generative model (GM) [R#1, R#2]:** We consider the setup with a GM in this work, which is common in theoretical
RL literature. Our primary aim is to argue the utility of ME in exploiting low-rank structures for RL. By assuming
a GM, we focus on providing key insights of the framework, while offering comparisons with structure-oblivious
literature. However, we strongly believe our proposal is more broadly applicable, e.g., to the online setup. Indeed, we
can still apply the "sample and pseudo-explore (via ME)" scheme with appropriate modifications. The most prominent
challenge anticipated in online setup is that we are no longer able to sample "any" state-action pair freely and adaptively;
the sampling needs to respect the exploration policy. This also implies that a more refined ME method needs to be
designed to handle the difficulties caused by diminished sampling capability. We believe our structured RL can bring a
similar complexity gain in general setups, and hope this can motivate further research in both RL and ME communities.

**2. Discount factor [R#3, R#4]:** It is true that our analysis requires $\gamma$ to be small. We would like to clarify that the
requirement stems from our analysis of ME method, and it does not necessarily indicate the limitation of the proposed
framework. As a matter of fact, we used large values of $\gamma$ in our experiments to give evidence that our algorithm can
be effective beyond the range of $\gamma$ allowed in analysis. The constraint on $\gamma$ arises from requiring $\|Q^{(t)} - Q^*\|_\infty$ to
decrease in the proof of Theorem 2. Our algorithm combines one-step lookahead and ME to update $Q^{(t)}$; the ME step
"amplifies" the error by a factor of at most $c_{me}$ with Assumption 1. In the end, $\gamma c_{me} < 1/2$ is required. However, we
believe it is an artifact of the conservative nature of our decoupled analysis. Indeed, there are several ways to relax the
restriction, e.g., by improving analysis to achieve a better $\ell_\infty$ guarantee or by devising novel ME methods for RL.

**3. Anchor points [R#1, R#4]:** Our proposed ME method relies on the existence and availability of a set of anchor
states/actions. The existence of an anchor set is straightforward from linear algebra. Viewing $Q^*(\mathcal{S}, \mathcal{A})$ as a (possibly
infinite-sized) matrix of rank $r$, there exists $\mathcal{S}^\sharp \subset \mathcal{S}$ s.t. $\{Q^*(s, \mathcal{A}) : s \in \mathcal{S}^\sharp\}$ spans the row space of $Q^*(\mathcal{S}, \mathcal{A})$ (likewise,
$\exists \mathcal{A}^\sharp$ spanning the column space of $Q^*(\mathcal{S}^\sharp, \mathcal{A})$). Now there remains the algorithmic question. We did not discuss this
point in the paper to avoid digression to secondary details, but we believe it won't be hard to find $\mathcal{S}^\sharp, \mathcal{A}^\sharp$ under mild
assumptions. For example, if the principal components of $Q^*$ are "incoherent" (i.e., $\|f_i\|_\infty/\|f_i\|_2$ and $\|g_i\|_\infty/\|g_i\|_2$
are small for $i \leq r$) and there is a sufficient "eigengap" ($\sigma_r$ is well separated from 0), then a random sample of sufficient
size would yield anchor sets with high probability, cf. discussions in Appendix G.2. Lastly, we remark that the anchor
selection needs not be perfect in practice, due to the robustness implied by our results for approximate rank-$r$ setup.

**[Reviewer #1] Warm-up example.** First, we would like to clarify that this is only a toy example meant to develop
readers' intuition with elementary analysis. Indeed, positive reward is not really needed for our general results and in
this toy case, one might also consider shifting the rewards by adding a constant to ensure $c_{me}$ is not prohibitively large.

**Sample complexity.** The goal of this work is to study the sample complexity for RL with continuous $\mathcal{S}, \mathcal{A}$, and to
understand if the dependence on the "size" of $\mathcal{S}$ and $\mathcal{A}$ can be improved by exploiting the structure. As the dependence
on $\gamma$ is of secondary interest to us, we treat $\gamma$ as a constant and hide it in the big-$O$ notation. We included the dependence
on $\gamma$ of the results from [3], [35], [36] in Table 1 for complete reference, however, we can omit them to avoid confusion.

**[Reviewer #2] Interpolation methods.** Note that we balance three error terms arising from Steps 2, 3, 4 of our RL-ME
algorithm in the proof of Theorem 2 by choosing parameters appropriately. In this work, we only assume Lipschitz
smoothness of $Q^*$; thus we don't expect order-wise gain from other interpolation methods and the eventual $\beta^{(t)}$-net
needs to be as fine as $O(\epsilon)$. However, it could be beneficial to use a more refined method, e.g., local polynomial
interpolation, if we assume higher-order smoothness or a parametric family of $Q$ functions, such as neural networks.

**Experimental section.** We agree, and it was mainly due to the limited space. For
the computational costs, nuclear norm minimization is known to be expensive for
large matrices. Here we show our computational benefits: we summarize the ME

| ME | Soft-Imp. | Nuc. norm | **Ours** |
|---|---|---|---|
| Runtime (s) | $41.5 \pm 1.7$ | $76.3 \pm 8.2$ | $\mathbf{1.9 \pm .6}$ |

runtime for Inverted Pendulum with a $2500 \times 1000$ matrix at one iteration. We will add related results in our revision.

**[Reviewer #3] Experiments.** This work is primarily focused on theoretical introduction of the novel, powerful low-rank
RL framework. While we validate it via classical tasks, we surely welcome suggestions for more practical examples.

**Lower bound.** This is achieved by viewing $Q$-function as a $(d_1 + d_2)$-dimensional function and interpreting the
estimation of $Q^*$ as a non-parametric regression problem. Classical minimax lower bound (e.g., Sec. 2.6 of [44]) would
imply the result. Indeed, the lower bound for continuous $\mathcal{S}$ & finite $\mathcal{A}$ from [33] was obtained in the same manner.

**[Reviewer #4] Nuclear norm minimization (NNM).** Although NNM works well in practice and has been extensively
studied, there is no satisfactory $\ell_\infty$ guarantee for it known so far. As we consider "provable" sample efficiency, we need
a new ME method. Moreover, NNM is computationally expensive; the table above demonstrates our method is **40x**
faster than NNM. Please find further discussions on the "failure" of existing ME methods in Appendix G.2. We believe
these already highlight our contributions and address why technical advances for ME are needed in combining with RL.

[Meta-Review · NeurIPS 2020]

The contributions were unanimously appreciated (the paper introduces an interesting structure, the regret analysis including the low-matrix estimation part is interesting). We recommend the paper for acceptance and encourage the authors to account for the reviewers’ comments when preparing the camera-ready version of the paper.